# Late Pleistocene glacial terminations accelerated by proglacial lakes

Meike D. W. Scherrenberg[1], Constantijn J. Berends[1], Roderik S.W. van de Wal[1,2]

[1]Institute for Marine and Atmospheric research Utrecht, Utrecht University, 3584 CC Utrecht, the Netherlands
[2]Faculty of Geosciences, Department of Physical Geography, Utrecht University, Utrecht, the Netherlands

*Correspondence to*: M.D.W. Scherrenberg (M.D.W.Scherrenberg@uu.nl)

**Abstract.** During the glacial cycles of the past 800 thousand years, Eurasia and North America were periodically covered by large ice sheets causing up to 100 meters of sea level change. While the Late Pleistocene glacial cycles typically lasted 80 – 120 thousand years, the termination phases only took 10 thousand years to complete. During these glacial terminations, the North American and Eurasian ice sheets retreated which created large proglacial lakes in front of the ice sheet margin.

Proglacial lakes accelerate the deglaciation as they can facilitate ice shelves in the southern margins of the North American and the Eurasian ice sheets. These ice shelves are characterized by basal melting, low surface elevations and negligible friction at the base. Here we use an ice-sheet model to quantify the (combined) effects of proglacial lakes on the Late Pleistocene glacial terminations through their interplay with glacial isostatic adjustment (GIA) and basal sliding. We find that proglacial lakes accelerate the deglaciation of the ice sheets mainly because of the absence of basal friction underneath ice shelves. If the

friction underneath grounded ice is applied to floating ice, we find that full deglaciation is postponed by a few millennia, more ice remains during interglacial periods, and no extensive ice shelves are formed. Additionally, the large uncertainty in melt rates underneath lacustrine ice shelves translates to an uncertainty in the timing of the termination of a millennium at most.

Proglacial lakes are created by the depression in the landscape that remains after the ice sheet has retreated. The depth, size and timing of proglacial lakes depend on the bedrock rebound. We find that if the bedrock rebounds within a few centuries,

instead of a few millennia, the mass loss rate of the ice sheet is substantially reduced. This is because fast bedrock rebound prevents the formation of extensive proglacial lakes. Additionally, a decrease in ice thickness is partly compensated by the faster bedrock rebound, resulting in a higher surface elevation with lower temperatures and higher surface mass balance delaying deglaciation. We find that a very long bedrock relaxation time does not affect terminations substantially, but may lead to a later inception of the next glacial period. This is because inception regions, such as North-Western Canada, remain

below sea level throughout the preceding interglacial period.

## 1 Introduction

From paleoglaciology we can learn which processes are important for the evolution of ice sheets. This can improve our understanding of the response of the Antarctic and Greenland ice sheets under future warming. During the Late Pleistocene (~800 – 10 thousand years (kyr) ago), the North American and Eurasian continents were recurrently covered by large ice sheets

(e.g., Hughes et al., 2015; Batchelor et al., 2019). While a single glacial cycle took on average 80 – 120 kyr, their decay phases

only took 10 kyr. The climate underwent global-scale changes during these glacial terminations and sea levels increased by up to 130 meters (Lambeck et al., 2014; Clark and Tarasov, 2014; Simms et al., 2019) mostly due to mass loss of the major ice sheets. As a consequence, the planetary albedo decreased due to the smaller extent of snow, sea ice and ice sheets (Abe-Ouchi et al., 2013; Stap et al., 2014). Large volumes of carbon stored in the deep ocean were released (e.g, Denton et al., 2010;

Menviel, 2019; Hasenfratz et al., 2019; Sigman et al., 2021) which contributed to an increase in $CO_2$ concentrations by $80 - 100$ parts per million during interglacial periods (Bereiter et al., 2015). These processes and changes in insolation, which are an important pacer for glacial cycles (Milankovitch, 1941), caused global temperatures to increase by roughly $4 - 5$ °C (Annan et al., 2022). While each of these processes enhanced the mass loss of the ice sheets, glaciated regions also affect the climate and deglaciation. It has for instance been suggested that the Late Pleistocene deglaciation phases only take place if the ice

sheets are large enough (e.g., Abe-Ouchi et al., 2013; Berends et al., 2021; Parrenin and Paillard, 2003). Additionally, Berends et al. (2021) suggested that this may lead to three ice regimes: a small ice sheet that melts every orbital maximum (e.g., Eurasia, Early-Pleistocene North America), a medium sized that survives orbital maxima through elevation-temperature and albedo-temperature feedbacks (e.g., interstadial North America) and a large ice sheet regime which, due to bedrock-ice feedbacks, becomes sensitive to small increases in summer insolation (e.g. North America at glacial terminations). North America may

have undergone a change in size regime during the Mid Pleistocene Transition (MPT; 1.2 Ma – 0.8 Ma), where glacial cycle periodicity shifted from 40 kyr to an average 100 kyr. This shift may have been caused by the removal of regolith during consecutive glacial cycles (e.g., Clark and Pollard, 1998; Tabor and Poulsen, 2016; Willeit et al., 2019), exposing the bedrock underneath and reducing sliding. This results in thicker ice sheets that are less sensitive to insolation maxima and can survive some insolation optima.

There are various processes controlling the interactions between the ice sheets and the climate and vice versa. Regions with ice and snow have a high albedo, which increases the amount of solar irradiance that is reflected and decreases global and local temperatures. An ice albedo-feedback, where albedo decreases due to the retreat of the ice sheets amplifies temperature increase and ablation. Additionally, the elevation of an ice sheet influences the surface mass balance, inducing a positive surface mass balance height feedback. A decrease in surface elevation increases temperatures which enhances ablation and

causes a further decrease in ice thickness. However, the ice sheet mass balance also depends on the amount of accumulation. The amount of precipitation may decline with elevation as decreasing temperatures lower the vapour pressure and limit the available moisture content this leads to a negative feedback. At the same time orographic forcing of precipitation can result in windward and leeward effects, depending on both ice-sheet geometry and prevailing winds. The ice-sheet topography can also influence large-scale atmospheric circulation (Beghin et al., 2015; Löfverström et al., 2016), influencing both temperature and

accumulation patterns (Pausata et al., 2011; Ullman et al., 2014; Liakka et al., 2016) on a global scale. Besides melting at the surface, ice sheets can also lose mass at the base. Specifically, water underneath ice shelves can facilitate melting, which can thin ice shelves, thereby reducing buttressing and accelerating ice mass loss. Additionally, mass is also lost at the margin of ocean or lake terminating ice sheets due to calving.

Besides these changes in the forcing, dynamical processes in the ice sheet can also influence the mass loss rates. Marine ice sheets where the grounding line rests on a retrograde slope may exhibit an instability, where a small perturbation can cause a self-sustained advance or retreat (Weertman, 1974; Schoof, 2012). This process is referred to as marine ice sheet instability (MISI) and it is thought to be especially important for the current mass loss in the West-Antarctic ice sheet (Pattyn, 2018), which has substantial parts of its grounding line resting on retrograde slopes. In the past decade, many improvements have been made in capturing MISI in ice-sheet models (e.g, Pattyn et al., 2012, 2013; Schoof, 2007; Schoof, 2012; Sun et al., 2020).

The North American and Eurasian ice sheets may have undergone the lacustrine equivalent of MISI during deglacial phases, called proglacial lake ice sheet instability (PLISI; Quiquet et al., 2021; Hinck et al., 2022). Proglacial lakes are created by the combination of glacial isostatic adjustment (GIA; Peltier, 1974) and runoff. The large mass of the ice sheet prompts bedrock deformation which creates a depression in the landscape. As the ice sheet starts to retreat, the rebound lags behind in time, creating an ice-free depression in front of the ice margin. This depression can fill up with melt water, creating a proglacial lake. Evidence for the existence of large proglacial lakes during the last deglaciation has been found in North America (Lake Agassiz; Upham, 1880; Lepper et al., 2013) and Eurasia (Baltic ice lake; Patton et al., 2017). Besides PLISI, proglacial lakes facilitate the same mass loss processes as marine shelves, though the rates may be different. Lake calving rates are typically at least one magnitude smaller compared to tide-water glaciers (e.g., Warren et al., 1995; Warren and Kirkbride, 2003; Benn et al., 2007). Sub-shelf melting is also different mainly due to the lack of salinity gradients driving sub-shelf circulation under ocean conditions (Sugiyama et al., 2016).

The interaction between North America and proglacial lakes has been studied using numerical models (e.g., Tarasov and Peltier, 2006; Hinck et al., 2022; Quiquet et al., 2021; Austermann et al., 2022). Hinck et al. (2022) and Quiquet et al. (2021) studied the effect of PLISI on the deglaciation of North America. They showed that proglacial lakes significantly accelerate the melt of the ice sheet, with the PLISI-induced mass loss being accelerated by the increased surface melt rates over the low-lying lacustrine shelves. Both studies find that the enhanced retreat by proglacial lakes is not caused by calving or basal melting, but rather due to PLISI and the negative surface mass balance. This is because ice shelves have a low surface elevation with high temperatures and strong ablation.

Here, we expand on the work by Hinck et al. (2022) and Quiquet et al. (2021) by considering a wider range of processes related to the presence of proglacial lakes during the deglaciation. We do this by using an ice-sheet model that includes both the North American and Eurasian ice sheets on consecutive glacial cycles. We use an ice-sheet model forced by General Circulation Model climate time-slices to study the transient effect of proglacial lakes on glacial terminations throughout the Late Pleistocene. Rather than making detailed sea level projections, our main goal is to investigate ice dynamical processes that may have contributed to the melt of the North American and Eurasian ice sheets. We focus on the effects of basal sliding, shelf formation, and sub-shelf melting on glacial terminations. In addition, we consider the effect of different GIA response time scales on proglacial lakes and glacial terminations.

## 2. Methods

### 2.1 Ice-sheet model

We simulated the Northern Hemisphere ice sheets using the vertically integrated ice-sheet model IMAU-ICE version 2.0 (Berends et al., 2022). The hybrid shallow ice / shallow shelf approximation is used to calculate the flow of ice (Bueler and Brown, 2009). To model GIA, we use an elastic lithosphere relaxing asthenosphere model (ELRA; Le Meur and Huybrechts, 1996). Basal friction is calculated using a Budd-type sliding law (Bueler and van Pelt, 2015; see Appendix A). We include a simple basal hydrology scheme following Martin et al. (2011), in which pore water pressure depends on ice thickness and the bedrock height compared to sea level. The till friction angle in North America is determined using the geology map from Gowan et al. (2019), which was specifically created for ice sheet model applications. For the Eurasian ice sheet, we generated a till friction angle map based on the sediment thickness map from Laske and Masters (1997), where we use till friction angles of 30 degrees for sediment thicknesses below 100 meters and 10 degrees for thicknesses exceeding that threshold. The 100 m threshold is large, but due to the coarse resolution of the Laske and Masters (1997), this will not impact the simulations substantially. Basal friction at the grounding-line is treated by a sub-grid friction-scaling scheme (Berends et al., 2022) and is based on the approach used in the Community Ice Sheet Model (CISM; Leguy et al., 2021) and the Parallel Ice Sheet Model (PISM; Feldmann et al., 2014). In this method, the basal friction coefficient of each model grid cell is multiplied with the grounded fraction of that cell. Therefore, friction is decreased where ice is partially floating (grounding line), and is negligible where ice is fully floating (shelves). This grounded fraction is calculated using the approach by Leguy et al. (2021) by bilinearly interpolating the thickness above floatation. Berends et al. (2022) show that the model performs well in the MISMIP and MISMIP+ benchmark experiments, resolving the (migrating) grounding line to within a single grid cell across a range of resolutions.

Calving is parameterised by a simple thickness threshold scheme, using a threshold thickness of 200 meters. To calculate sub-shelf melt, we use a depth-dependent sub-shelf melt parameterization (Martin et al. 2011) where the basal melt is linearly related to the anomaly of the temperature. Ocean temperatures are parameterised, with globally uniform ocean temperature changes (De Boer et al., 2013) which do not capture regional variations in ocean temperatures. We apply the same sub-shelf melting method for oceans and proglacial lakes unless stated otherwise. This is a simplification, as the lacustrine and marine environment have different thermal structures, salinity and sub shelf circulation regimes (Sugiyama et al., 2016). Lakes and ocean are simulated when the bedrock is below the modelled sea level. Sea level is calculated while maintaining a constant ocean-area and only accounting for ice volume above floatation. As the ice sheet grows, the bedrock subsides due to GIA. At glacial maxima, large areas of the ice sheet can thus become grounded below sea level. During a deglaciation, the ice sheet retreats and thins, but the bedrock uplift lags the changes in ice load. If the bedrock is below sea level, the cell becomes one of three different types of surfaces: (1) The ice is thick enough that it is grounded and the cell is considered as grounded ice. (2) The ice is thin enough that it can float, and it becomes an ice shelf. (3) There is no ice, and this cell is considered as ocean. There is no distinction between lake and ocean in the model, and processes such as calving and basal melt are treated the same

for both, indicating that these processes may be overestimated in the lakes. A further simplification is the constant surface level of the lakes, the consequences of this is explored in paragraph 3.5.

The North American, Eurasian and Greenland ice sheets are simulated in three separate domains. North America and Eurasia have a 40 x 40 km spatial resolution and Greenland 20 x 20 km. The boundaries of these domains are shown in Fig. 1. The higher resolution of the Greenland ice sheet results in a similar number of grid-cell compared to the other two domains,

while capturing smaller topographic features. As shown in Fig. 1, the domains have some overlapping regions. Therefore, regions that appear in more than one model domain are only allowed to have ice in one of them, e.g., ice on Ellesmere Island is only simulated in the North American domain but not in the Greenland domain, while ice on Greenland itself is not simulated in the North America domain. We simulate Greenland and North America in separate domains, but they are thought to have merged during glacial periods.

**2.2 Climate forcing**

To calculate the melt and accumulation of ice, our surface mass balance model (see section 2.4) requires information on precipitation and temperature as a function of time and space. To obtain the climate forcing computationally efficiently we interpolate between pre-calculated pre-industrial (PI) and last glacial maximum (LGM; 21 kyr ago) time-slices using a matrix method (Pollard et al., 2010). This allows us to implicitly include climate – ice-sheet interactions at low computational costs

compared to fully-coupled ice-climate set-ups. Details of this method, which is based on Berends et al. (2018) and Scherrenberg et al. (2023), are described in appendix B.

The main driver of temperature change is the external forcing and the albedo, both contriving 50% to the final temperature interpolation. External forcing is shown in Fig. 2a and is a combination between $CO_2$ (Fig. 2b; Bereiter et al., 2015) and insolation change (Fig. 2c; Laskar et al., 2004). To derive an interpolation weight from this forcing, we first

determine an index for $CO_2$, where 0 is LGM (190 ppm) and 1 is PI (280 ppm) climate. We then modify this index using the 65°N summer insolation to capture changes in temperature caused by the orbital cycles. Therefore, when summer insolation decreases the climate forcing becomes closer to LGM leading to cooling. The forcing index remains unchanged if the insolation is 440 W/m$^2$ (see Fig. 2a and Fig. 2c). As a result, for LGM $CO_2$ concentrations, the forcing index can still be relatively high for strong insolation, and the forcing index for PI $CO_2$ levels can be relatively low for weak insolation values. We have tuned

the effect of insolation on the forcing index to capture the glacial cycle periodicity across the past 800 thousand years. To calculate the albedo feedback, we multiply the modelled surface albedo with the insolation to obtain the annual amount of insolation absorbed by the surface. We then calculate an interpolation weight from the concurrent amount of absorbed insolation and compare it to the fields obtained with LGM and PI climates and masks. The matrix method includes a precipitation-topography feedback as precipitation is interpolated with respect to the local and domain-wide change in

topography relative to the LGM and PI topography.

## 2.3 Climate time-slices and downscaling

By simulating the last glacial cycle using an ice-sheet model, it has been shown that the LGM extent and volume are strongly dependent on the climate forcing (Charbit et al., 2007; Niu et al., 2019, Alder and Hostetler, 2019; Scherrenberg et al., 2023). Not all GCM simulations can be used to model an LGM ice sheet extent that agrees well with reconstructions (Scherrenberg et al., 2023; Niu et al., 2019). Here we use the mean of MIROC (Sueyoshi et al., 2013), IPSL (Dufresne et al., 2013), COSMOS (Budich et al., 2010) and MPI (Jungclaus et al., 2012) members of the paleoclimate modelling intercomparison project phase 3 (PMIP3; Braconnot et al., 2011). This ensemble has been shown to yield good LGM extent in combination with IMAU-ICE (Scherrenberg et al., 2023). To correct for biases in the GCM data, we calculate the difference between the PI time-slice and the reanalysis from ERA40 (Uppala et al., 2005). This bias is then applied to both the PI and LGM time-slice. As a consequence, the resulting PI time-slice may contain some of the anthropogenic warming enclosed in ERA40.

The topography and spatial resolution differ between the climate forcing and the ice-sheet model. Therefore, some corrections need to be applied before the climate forcing can be used in IMAU-ICE. First, we bilinearly interpolate the climate forcing to the finer ice-sheet model grid. As the climate forcing has a lower resolution and therefore a smoother topography, some topographic corrections need to be applied to the temperature and precipitation fields. For temperature, we apply a lapse-rate-based correction. For precipitation we use the Roe and Lindzen (2001) model to capture the orographic forcing of precipitation on the sloping ice margin, and the plateau desert effect in the ice-sheet interior. A more detailed description of the bias correction and downscaling methods can be found in appendix C.

## 2.4 Surface mass balance model

The surface mass balance (SMB) is calculated monthly using IMAU-ITM (insolation-temperature model; Berends et al., 2018). For the present-day climate this provides an adequate SMB distribution as shown in the Greenland surface mass balance model intercomparison project (GrSMBMIP; Fettweis et al., 2020). Using this model, accumulation of snow is calculated using the large-scale snow-rain partitioning proposed by Ohmura, (1999). Refreezing is calculated following a scheme by Huybrechts and de Wolde, (1999) and Janssen and Huybrechts, (2000). Ablation is calculated based on Bintanja et al. (2002) and depends on temperature, insolation and albedo. The equations describing this scheme (IMAU-ITM) are discussed in more detail in appendix D.

## 3. Results

We conducted a "Baseline" by simulating the North American ice sheets from 782 ka to present-day. Fig. 3a shows the extent of the ice sheets at the LGM compared to a reconstruction of North America by Dalton et al. (2020) and Eurasia by Hughes et al. (2015; orange contours). The modelled LGM extent matches the reconstructions reasonably well, though ice coverage is lacking in the British island. A small proglacial lake is created in North America around 14 kyr ago (Fig. 3d). The North American ice sheet retreats faster compared to the reconstruction from 12 kyr ago onwards (Fig. 3e-h), with full retreat already

reached at 10 kyr ago rather than three to four millennia later. Amongst other reasons (e.g., uncertainty in climate forcing, atmospheric circulation, ice-sheet model parameterizations), this discrepancy could be partially due to the absence of a feedback between melt water, the ocean and climate. For example, prior to the Younger Dryas (12.9-11.7 kyr ago), large amounts of fresh water would flow into the North Atlantic (Teller et al., 2002; Tarasov and Peltier, 2005, 2006; Condron and Winsor, 2012), which can collapse the Atlantic meridional overturning circulation (AMOC) and would result in cooling (McManus et al., 2004; Velay-Vitow et al., 2024) and a stagnation in sea level rise (Lambeck et al., 2014). At present-day (see Fig. 3i), the North American and Eurasian ice sheets have fully melted, except for some small ice caps in regions that are currently partly glaciated (e.g., Artic Archipelago, Svalbard, Iceland).

The total sea-level contribution of this Baseline simulation is shown in Fig. 4 and compared to ice volume reconstructions by Spratt and Lisiecki (2016), and Grant et al. (2014). Since we only simulate Northern-Hemisphere ice sheets, we added 30% to the ice sheet contribution to account for sea level changes caused by processes other than the Northern-Hemisphere ice sheets, such as the Antarctic (~10 m), Greenland and Patagonian ice sheets (Simms et al., 2019). While Antarctic ice volume was shown to be correlated to Northern Hemisphere ice volume as sea level change prompts grounding line advance and retreat (Gomez et al., 2020), this approach neglects out of phase behaviour between Northern and Southern Hemisphere ice mass loss.

We find that the modelled sea level captures all major melting events except MIS 13 (marine isotope stage; 530-480 kyr ago), which has relatively low atmospheric $CO_2$ concentrations (~240 - 250 ppm) compared to the interglacial periods that succeeded it (Siegenthaler et al., 2005; Bereiter et al., 2015). The modelled inception periods are long compared to reconstructions, especially the warm interglacial MIS 11 (425-375 kyr ago). This results from the bias correction based on observations where our PI time-slice shows some anthropogenic warming. Consequently, modelled ice inception requires relatively low $CO_2$ concentrations and weak insolation. The ability of the model to capture the overall pattern of glacial terminations and fully melting North American and Eurasian ice sheets allows us to study the importance of ice dynamical processes that may have contributed to the decay of the ice sheets.

Fig. 5 shows the ice volume time-series of North America (Fig. 5b), Eurasia (Fig. 5c) and the Northern Hemisphere (Fig. 5a), colours indicating net melt (red), net accumulation (blue), and red points indicating the onset of deglaciations. In Fig. 5d-f, the ice volume is plotted against climate forcing, with a glacial climate on the left and interglacial climate on the right. When the climate becomes colder the ice sheet will tend to have a net positive mass balance and will grow. However, when the ice sheet becomes larger and grows more towards the warmer south, the $CO_2$ concentration and insolation need to be lower to be able to maintain net growth. This agrees with Abe-Ouchi et al. (2013) and Parrenin and Paillard (2003) and shows that a larger ice sheet will be more vulnerable to increases in insolation, and that deglaciations tend to take place when ice volume is large enough.

The Eurasian ice sheet is more sensitive to collapse compared to the North American ice sheet. Occasionally, the Eurasian ice sheet melts completely (e.g.,168 kyr ago, 50 kyr ago), while the North American ice sheet only undergoes partial melt. This suggest that the Eurasian ice sheet is more vulnerable to complete melt during climate optima and that $CO_2$

concentrations and insolation can facilitate decay of the Eurasian ice sheet and at the same time be favourable enough for the North American ice sheet to survive an interglacial. On average, Eurasia achieves full deglaciation roughly three millennia earlier compared to the North American ice sheet.

This is in line with Bonelli et al. (2009), Abe-Ouchi et al. (2013), and Tarasov and Peltier (1997b), finding that the Eurasian ice sheet needs lower $CO_2$ concentrations or insolation compared to the North American ice sheet to survive climatic optima. The higher sensitivity of the Eurasian ice sheet also follows from ice reconstructions, such as Gowan et al. (2021) and Mangerud et al. (2023), who show that the Eurasian ice sheet lost most of its volume during the MIS 3 (60-25 kyr ago) interstadial, while the North American ice sheet continued to survive until the LGM. This higher sensitivity for Eurasia to melt during a climate optimum may in part be due to the smaller size: Eurasian sea level contribution at LGM is 19.1 m, compared to 80.1 m for North America, and higher LGM temperatures in the climate forcing, resulting largely from the thinner Eurasian ice sheet.

## 3.1 Design of the perturbed experiments

To investigate the effect of proglacial lakes and GIA on the Late Pleistocene terminations, we carry out a set of experiments that are similar to the Baseline experiment, but have one process changed at a time. In the Baseline set-up, our model reproduces the basic features of glacial terminations throughout the Late-Pleistocene. For the sensitivity experiments, we modify the Baseline simulation to investigate the effect of sub-shelf melting, basal friction of grounded ice, sub-shelf basal friction and GIA on the deglaciation of the Northern Hemisphere ice sheets. Each simulation branches off from the Baseline simulation at 782 kyr ago, which is during an interglacial period when the North American and Eurasian continents were mostly ice-free. In the next few sections, we introduce these perturbation experiments to investigate which processes are important for the decay of the ice sheets. These experiments are described in paragraphs 3.2-3.6 and summarized in Table 1.

**Table 1:** A description of the experiments. Each perturbed experiment is similar to the Baseline except for the described feature.

| Experiment | Description | Section |
| --- | --- | --- |
| Low Friction | Till friction angle is set to 10 degrees, representing full sediment coverage | 3.2 |
| High Friction | Till friction angle is set to 30 degrees, representing full bedrock coverage | 3.2 |
| Zero BMB | Basal mass balance is set to 0 everywhere | 3.3 |
| Rough Water | The basal friction of floating ice is the same as land | 3.4 |
| Lake 100m | Increased lake levels in North America by 100 m | 3.5 |
| Lake 200m | Increased lake levels in North America by 200 m | 3.5 |
| Lake 300m | Increased lake levels in North America by 300 m | 3.5 |
| Lake 400m | Increased lake levels in North America by 400 m | 3.5 |
| Fast GIA | GIA relaxation time of 300 years | 3.6 |

**3.2 The effect of basal friction of grounded ice**

The till friction angle is prescribed and based on the geological map (Gowan et al., 2019; North America) and sediment map
(Laske and Masters, 1997; Eurasia). To assess the impact of basal friction on the deglaciation, we conducted two sensitivity
experiments. First, the Low Friction simulation where till friction angles are set to 10 degrees over the entire North American
and Eurasian domains and is a rough representation of continents fully covered by sediments, which are easily deformed.
Secondly, the High Friction simulation roughly represents full bedrock coverage where the till friction angles are set to 30
degrees. The basal hydrology is applied the same for all till friction angle maps and does not distinguish between sediment or
bedrock.

Fig. 6 shows a time-series of the global mean sea level contribution for the Low Friction, Baseline and High Friction
experiments. Panels b-k zoom in on individual glacial terminations. The basal friction has a substantial impact on the
deglaciation. As shown in Fig. 6, decreasing till friction angle always results in an earlier completed deglaciation. In the Low
Friction simulation, the peak melt rate in North America is achieved an average four centuries earlier compared to the Baseline,
while for the High Friction simulation, the peak mass loss rate is achieved a millennium later (see Fig. S1, S2). In the Low
Friction scenario, the ice sheet does fully melt at MIS 13, while in the High Friction full deglaciation is not reached at either
MIS 13 and MIS 17 (712 – 676 kyr ago).

Increasing friction will also lead to increased ice volume at glacial maxima, though ice extent is not impacted as
much. At the LGM, North American global mean seal level contribution in High Friction is 93.6 m (17% larger) compared to
the Baseline (80.1 m), while Low Friction has a contribution of 75.3 m (6% smaller). Similarly, the Eurasian sea level
contribution at LGM is 19.1 m for the Baseline, 18.8 m for Low Friction and 23.7 m for High Friction. Fig. 7, shows maps
with ice thickness for the Baseline, Low Friction and High Friction. Ice area at the LGM (Fig.7 a,f,k) deviates less than 1%
for North America and 3% for Eurasia between each simulation. Additional ice volume therefore mostly results from the
increase in thickness (see Fig.7a,f,k). A time-series comparing ice volume and area over the entire Late Pleistocene is shown
in Fig. S3.

These results suggest that basal friction has a large influence on melt rates during climate optima. Lower friction
results in thinner ice sheet with gentler slopes, a large ablation area and lower SMB. Combined with increased ice velocities
transporting more ice towards the ablation area, these processes can explain the increased sensitivity of the ice sheet during
deglaciations and interstadial periods. As friction is reduced, the ice sheet may become more sensitive to collapse during
climate optima, which is at the base of the regolith hypothesis, where the MPT may be explained by an increase in basal drag
as sediment has been gradually removed during glacial cycles.

To study the robustness of our results, we also conducted the Low Friction and High Friction experiments with the
set-up used for the Rough Water and Fast GIA simulations discussed in section 3.4 and 3.6 (see Fig. S4, S5).

### 3.3 The effect of basal melt on glacial cycles

Proglacial lakes facilitate ice shelves, which can undergo sub-shelf melting. Sub-shelf melting is considered an important process for the current mass-loss of Antarctica (Pritchard et al., 2012; Shepherd et al., 2018), where sub-shelf melting is dominated by temperature and salinity gradients. Lake Agassiz was a fresh water lake created by the melting ice sheet and could therefore have substantially reduced basal melt rates compared to parameterizations made for ocean shelves. Here, we conduct a sensitivity test to investigate whether sub-shelf melting has a significant impact on the retreat of the North American

and Eurasian ice sheets.

The Zero BMB experiment deviates from the Baseline by setting the sub-shelf melt rate in North America and Eurasia to 0. Fig. 8 shows the ice-volume time-series calculated in the Zero BMB experiment and compares it to the Baseline. Fig. 8b-k show the terminations in more detail. Zero BMB is similar to the Baseline, though the ice sheets during glacial periods are slightly bigger and shelves can extent far into the ocean. During most terminations removing sub-shelf melting only has a

small effect, delaying full deglaciation by up to a millennium. MIS 13 and MIS 17 are exceptions. In MIS 17, substantially less ice is melted in the Zero BMB experiment compared to the Baseline, with lowest Northern Hemisphere sea level contribution reaching 15.9 m in the Zero BMB and 4.0 m in the Baseline. Contrarywise, more ice is lost during MIS 13 in the Zero BMB simulation, despite the lack of sub-shelf melting. This is caused by the increased sensitivity of the ice sheets when ice volume is higher, as is shown in the relationship between ice sheet decay and climate forcing in Fig. 5. The Zero BMB has

a substantially larger ice volume during the glacial period preceding MIS 13, and the retreat produces a substantial proglacial lake while the Baseline does not (see Fig. S6).

### 3.4 The effect of basal friction of floating ice

The ice shelves floating on proglacial lakes or seas experience negligible basal friction, which results in relatively high flow velocities. To study the impact of this lack of friction, we conduct the Rough Water experiment. In this experiment the sub-

grid friction coefficient is not scaled with the grounded fraction, but is instead calculated as if all grid cells are grounded. Therefore, friction under shelves is not negligible anymore. This essentially prevents the formation of shelves, so that a migration of the grounding line will not cause a change in friction, which prevents PLISI/MISI. While this is a very unrealistic scenario, the grounding line does not migrate far into the ocean to cover the entire ice domain due to strong ablation, which can be seen in the ice thickness and bedrock topography maps in Fig. 9i-l.

The sea level contribution for the Rough Water experiment shown in Fig. 8 can be compared to Zero BMB and the Baseline. During the onset of the termination, the Rough Water experiment losses mass at roughly the same pace as the Baseline. However, once more than half of the ice volume is lost, the mass loss rate in Rough Water slows down with respect to the Baseline. This is because the proglacial lakes are only created once the ice sheet has already partly retreated. Therefore, while the Baseline and Rough Water experiments have a similar retreat rate at the onset of the termination, the retreat in the

Baseline simulation accelerates once the proglacial lake has formed. Further differences between the Rough Water and

Baseline can be seen in the transect shown in Fig. 10. While the Baseline simulation has large ice shelves in North America, almost the entire ice sheet is grounded in the Rough Water simulation. The shelves in the Baseline have large ice velocities due to the negligible sub-shelf friction. Consequently, the surfaces of the ice shelves in the Baseline simulation are flat and close to sea level and therefore experience high temperatures and strongly negative SMB. In the Rough Water simulation, these shelves are very small. Without extensive shelves, the higher elevation and steeper slopes result in a smaller ablation area.

Interglacial ice volume is generally larger in the Rough Water simulation compared to the Baseline. Eurasia melts completely only during glacial terminations, rather than also during interstadial periods. Additionally, more ice tends to survive interglacial periods in both North America and Eurasia, which can be observed in the Barents-Kara Sea region and the Arctic Archipelago (see Fig. 9l). These results are sensitive to basal friction though, and an increase in friction can lead to more ice surviving through interstadial periods. In the High Friction equivalent to the Rough Water experiment neither Eurasia nor North American fully deglaciate, with smallest sea level contributions of 4.5 m and 11.5 m respectively (see Fig. S5).

## 3.5 Lake Depth

Lakes are assumed to exist in the model wherever the (GIA-adjusted) bedrock is below the modelled sea level, and the modelled ice is thin enough to be floating. In the Baseline experiment, the water level of the modelled lakes is assumed to be equal to sea level, while in reality this can be significantly higher. To assess the effect of this simplifying assumption on our results, we simulate the last deglaciation (21 kyr ago – present-day) with a set of four different constant sea levels and apply it to the entire ice-sheet model grid (Lake 100m, 200 m, 300 m and 400 m) relative to present day. Therefore, lakes are simulated when bedrock is below these fixed sea levels, resulting in increased lake levels. As these sea levels are applied to the entire ice-sheet model grid, the experiments shown here are therefore an overestimation of the melt rates compared to a high-resolution lake model which allows for more drainage. Obviously very high sea levels (e.g., 400 m) create an unrealistic inland sea that can hamper glacial inception, so instead we only apply these fixed levels during the last deglaciation and focus on the North American ice sheet only. Despite that, even our most extreme scenario (400 m) does not induce an immediate collapse after the start of the simulation.

A time-series of ice volume for the four Lake level experiments and the Baseline is shown in Fig. 11a. The evolution of ice extent is shown in Fig. 11b-f, colours indicating when each region of North America was last covered by ice. Here we find that the retreat of the North American ice sheet is much faster with increasing sea level, translating to roughly one millennium lead in terms of ice volume. These results indicate that lake levels are important to the rate of deglaciation for the North American ice sheet and that higher lake levels can substantially accelerate deglaciation.

## 3.6 Glacial isostatic adjustment

Proglacial lakes are created by the interaction between GIA, ice sheets and melt water, and as a consequence the GIA relaxation time controls how quickly the bedrock fully recovers from a change in ice load. This relaxation time, as well as the thickness

of the ice sheet and the retreat rate, control the size and shape of the proglacial lake. Here we assess the effect of GIA response time on multiple glacial cycles while modelling PLISI. We compare three simulations; the previously shown Baseline

simulation (3000-yr), the Slow GIA (10,000-yr), and Fast GIA (300-yr relaxation time).

Fig. 12 shows the ice volume time-series of these three simulations, with the smaller panels zooming in on individual terminations (Fig. 12b-k). The retreat in the Slow GIA simulation is generally slower and can lag up to a millennium behind the Baseline in terms of ice volume, though some deglaciations have minimal difference with the Baseline (e.g., the last and penultimate deglaciations). This could be explained by both difference in bedrock topography at the onset of deglaciations and

the proglacial lake. The slower subsidence rates in the Slow GIA simulations may result in higher bedrock topographies. This reduces the modelled saturation at the base, as saturation increases with decreasing bedrock topography relative to concurrent sea level. Therefore, basal friction may be lower in the Slow GIA at the onset of some terminations, resulting to reduced retreat rates. However, if the bedrock topography during a glacial maximum is similar to the Baseline, the retreat will be similar as well (see Fig. 12c,j,k). An exception is MIS 13, were the North American ice sheet in the Slow GIA fully melts while the

Baseline does not. Retreat rates and ice sheet volume are not large enough to induce a proglacial lake in the Baseline. However, in the Slow GIA simulation, the slower bedrock uplift allows for a proglacial lake to be created and sustained throughout this termination event, allowing the North American ice sheet to fully melt (see Fig. S6).

The ice thickness and bedrock topography for the Slow GIA simulation are shown in Fig. 9m-p for 21, 12, 10 and 0 kyr ago. The proglacial lakes during retreat are significantly larger compared to the Baseline, as the bedrock uplift is slower.

Slow GIA also has a delayed inception phase which can be seen from Fig. 12. Due to the slow bedrock uplift, large parts of North America are still below sea level millennia after the ice sheet fully receded (see Fig. 8o). As a result, regions such as North-Eastern Canada, a location for ice inception, are still below sea level throughout the entire preceding interglacial period.

Fast GIA has substantially slower melt rates compared to the Baseline, Slow GIA, and Rough Water simulations, which can be seen in transects shown in Fig. 10 and the ice thickness maps shown in Fig. 9q-t. The delayed deglaciation is due

to two processes. Firstly, as the ice sheet retreats, the rapid bedrock rebound quickly eliminates the depression left by the ice sheet, preventing the formation of proglacial lakes (See Fig. 9q-t). Secondly, the ice thickness decrease is more efficiently compensated by the bedrock rebound, reducing the elevation-temperature feedback and thereby reducing surface melt rates and slowing down the deglaciation. Therefore, while both the Fast GIA and Rough Water eliminate the effect of proglacial lakes, only the Fast GIA reduces melt rates from the onset of the termination. During many interglacial periods – including

present-day – an ice dome persists or the termination is skipped. However, this is again strongly dependent on the basal friction, with the Low Friction equivalent of the Fast GIA experiment inducing full melting during most interglacial periods, while the High Friction equivalent never melts completely (see Fig. S4, S5). The combination of the lack of lakes and the SMB-elevation feedback makes the Fast GIA the simulation with the slowest deglaciation.

## 4. Discussion

In this study, we investigate the effect of proglacial lakes on the deglaciation of the Eurasian and North American ice sheets throughout the past 800 kyr. In the Baseline configuration, the modelled ice volume over time generally agrees well with different sea-level reconstructions, so that all major deglaciations throughout the Late Pleistocene are captured. A shortcoming is the lack of ice coverage in the British Islands, and that our simulations tend to have too long interglacial periods compared to reconstructions, especially MIS 11 interglacial which had higher global temperatures and sea-levels compared to present

day (Hearty et al., 1999; Raymo and Mitrovica, 2012). Warmer than pre-industrial temperatures are also not captured, as our climate forcing was interpolated from PI and LGM time-slices. Adding additional time-slices to our matrix method, similar to Abe-Ouchi et al. (2013), particular for warmer than present-day conditions may improve our representation of different ice volume, $CO_2$ concentration and orbital parameters. Results are anyhow depending strongly on the quality of the climate forcing (Scherrenberg et al., 2023). Here we have chosen a forcing that would result in an LGM extent that agrees well with

reconstructions rather than a large number of time-slices.

The size and shape of proglacial lakes follow from the interaction between GIA and ice thickness. Here we have used uniform GIA response times, but in reality, GIA varies spatially (e.g., Forte et al., 2010; van Calcar et al., 2023) and has large uncertainties. GIA can increase the size of proglacial lakes, compared to no GIA (e.g., Austermann et al. 2022), and in this study we have shown that when increasing the GIA response time, proglacial lakes will be larger. A faster GIA response will

decrease the size of proglacial lakes, and the increased bedrock uplift can partly compensate thickness loss resulting in reduced melt rates.

Here we have simulated lakes when bedrock is below sea level. This is a simplification, as lakes can exist well above sea level, and as shown in the Lake 100 – 400 m experiments, increasing lake levels will result in a substantially faster deglaciation. Lake models generally have high computational costs, and to perfectly capture all valleys and drainage channels

they need to be applied at high resolution. Alternatively, it could be feasible to asynchronously couple a flood-fill algorithm to the ice-sheet model (see Berends et al., 2016), but these should also be applied at relatively coarse resolutions to maintain a reasonable computational time. While not perfect, in future research, a method such as this may provide a mid-way solution between computational resources and the quality of the lakes.

In this study we have treated lakes as if they were ocean, but due to the low salinity and smaller size of lakes they can

have substantially different sub-shelf melting (Sugiyama et al., 2016), lower calving rates (e.g., Warren et al., 1995; Warren and Kirkbride, 2003; Benn et al., 2007), and the clogging of icebergs can create a potential buttressing effect (Geirsdóttir et al., 2008). These limitations suggest that the effect of basal melt and calving may be overestimated in the proglacial lakes, and they are further enforced by uncertainties in the sub-shelf melting and calving schemes used here. For example, applying a more sophisticated sub-shelf melting scheme, which simulates high melt rates near the grounding line (Rignot and Jacobs,

2002), may result in a greater impact from sub-shelf melting. Additionally, here we have not explored the effect of calving rates in detail, but lake calving can have an effect on ice flow of the Laurentide ice sheet (e.g., Cutler et al., 2001). Nevertheless,

our results suggest that while sub-shelf melting can enhance melt of the North American and Eurasian ice sheet, it is overshadowed by large surface melt rates combined with PLISI. Due to these limitations, we cannot give an exact estimate to how much longer a deglaciation phase will take when removing sub-shelf melting, but it serves as an indication that full melt can take place without sub-shelf melting.

Another limitation is the treatment of basal friction. For basal hydrology we have used a parameterization from Martin et al. (2011), and till friction angle was based on geology and sediment masks. We performed experiments by decreasing the till friction angle to 10 or 30 degrees representing a full sediment cover or full hard bed coverage. We found that these have a substantial influence on the ice volume at glacial maxima, with a much smaller effect on the extent. Deglaciation is also slower with increasing friction. While our main conclusions are consistent regardless of basal friction, the timing of deglaciation and the ice volume that persists through interglacial periods differ substantially. This highlights that glacial cycle volume and melt rates are sensitive to the basal friction. Here we have used a static till friction angle mask. However, friction can change over time as bedrock is eroded and sediment is transported. Basal friction can therefore be improved by including a sediment transport model (e.g, Hildes et al., 2004; Melanson et al., 2013), combined with a more sophisticated basal hydrology method (e.g., Hoffman and Price, 2014; Flowers, 2015).

The matrix method, which we use to interpolate between the LGM and PI time-slices, implicitly includes a temperature-albedo and precipitation-topography feedback. However, ice-sheet climate interactions can exhibit threshold behaviours which cannot be simulated using our method. For example, the opening and closing of straits such as the Canadian Arctic Archipelago (Löfverström et al., 2022) or the response of Heinrich and Dansgaard/Oeschger events (Claussen et al., 2003), or rapid changes in ocean circulation and sea ice due to the influx of melt water into the ocean (Otto-Bliesner and Brady, 2010). Additionally, since we do not model the ocean, there are no interactions between melt water, ocean circulation and the climate. This may partially explain why the Baseline simulation has too high retreat rates from 12 kyr ago, coinciding with the Younger Dryas. The Younger Dryas has reduced rates of sea level rise (e.g., Lambeck et al., 2014) and lower Northern Hemisphere temperatures due to a stagnation of the AMOC (McManus et al., 2004; Velay-Vitow et al., 2024) which was likely caused by an influx of melt water into the North Atlantic (Teller et al., 2002; Tarasov and Peltier, 2005, 2006; Condron and Winsor, 2012). This process is not included in our set-up as temperature is only affected by $CO_2$, insolation, elevation and albedo, and therefore fail to capture the stagnation in melt rates during the Younger Dryas.

Including many of these behaviours would require a model that more explicitly simulates the climate system. GCM models may be able to simulate these interactions, but simulating glacial cycles with a reasonably high resolution would require an excessive amount of computational resources. Alternatively, ocean-atmosphere circulation models can be used to simulate individual glacial terminations (Obase et al., 2021), or intermediate complexity models (Ganopolski and Calov, 2011) can simulate multiple glacial cycles with more explicit feedbacks in the climate system, though still at higher computational costs than ice sheet models, and with more parameterizations compared to full GCMs.

## 5. Conclusion

We studied the relative importance of different ice-dynamical processes for glacial terminations. The onset of terminations is dominated by a decrease in SMB, which causes a retreat of the ice sheet. We found that the Eurasian ice sheet needs lower $CO_2$ concentrations and/or insolation compared to North America in order to survive a climate optimum. Once the ice sheets have retreated significantly, proglacial lakes are created at the margin of the ice sheets. Our results show that these proglacial lakes can significantly accelerate the collapse of the North American and Eurasian ice sheets. If certain processes facilitated

by the lakes are removed, the North America and Eurasia deglaciate at a reduced pace, and may remain partially ice-covered during interglacial periods.

The largest impact that proglacial lakes have on deglaciations is that they facilitate low friction under floating ice. If the basal friction of shelves is the same as grounded ice, which removes the effect of PLISI and MISI, the Eurasian ice sheet only fully melts during interglacial periods rather than interstadials, and more ice can persist through some interglacial periods.

The high ice velocities caused by the negligible sub-shelf friction creates large shelves with low surface elevations, which have high temperatures and surface melt. Though, this process is also sensitive to the basal friction. By lowering basal friction, ice thickness at glacial maxima is decreased and consequentially, surface temperatures and melt are increased. Therefore, a lower friction results into a faster deglaciation. This shows that modelling sediments and hydrology are important to simulate glacial cycles. Furthermore, we found that sub-shelf melting is only a secondary effect to the mass loss of the ice sheets. Applying a

zero sub-shelf melt rate still results in a full deglaciation, although it may take an additional millennium to complete.

We have also investigated the effect of the GIA response time on consecutive glacial cycles and deglaciations. If the GIA responds slower compared to our Baseline, the termination will be up to a millennium slower and the subsequent inception phase is delayed. Since the inception sites are typically also the last regions to deglaciate, the land can still be below sea level at the onset of the next glacial period only if the bedrock rebound is too slow. We find that a GIA response that is substantially

faster than the Baseline has a slower deglaciation and larger interglacial ice volumes as the North American and Eurasian ice sheet may not fully deglaciate during some interglacial periods. This is because proglacial lakes are not created when the bedrock uplift is too fast. Additionally, surface melt is reduced as the bedrock uplift more efficiently compensates the thickness loss.

The importance of understanding marine ice-sheet dynamics and ice-sheet climate interactions when projecting the

future mass loss of the Greenland and Antarctic ice sheets is thought to be very important. For Antarctica, the marine equivalent to PLISI, MISI may induce a run-away collapse of parts of West-Antarctica (e.g. Ritz et al., 2015). Though this is not a perfect analogy, as for example the high sensitivity of the Eurasian ice sheet towards temperature does not perfectly translate to future Western-Antarctic melt (van Aalderen et al., 2023). Greenland is currently mostly land terminating, but proglacial lakes are created during the retreat, which is already being observed (e.g., Carrivick and Quincey, 2014). These lakes may accelerate

the retreat of the Greenland ice sheet (Carrivick et al., 2022). Therefore, our results underline the fact that these processes are

just as relevant for understanding past ice-sheet evolution, so that reproducing this evolution can help constrain these processes in the context of current changes in Antarctica and Greenland.

*Code availability:* The source code for IMAU-ICE can be found at https://github.com/IMAU-paleo/IMAU-ICE. The version used in this study as well as the configuration files are available at Zenodo [DOI will be added upon acceptation]. Running simulations requires additional files for $CO_2$ (see Bereiter et al., 2015), climate (PMIP3 database: https://esgf-node.ipsl.upmc.fr/search/cmip5-ipsl/; last access 24 nov 2023), insolation (Laskar et al., 2004) and initial topography (ETOPO; Amante and Eakins, 2009; Bedmachine; https://nsidc.org/data/idbmg4/versions/1; last access 9 feb 2024). LGM land and ice masks were obtained from Abe-Ouchi et al. (2015). Till friction angle was obtained from Gowan et al. (2019) and Laske and Masters (1997). For more information, contact the corresponding author.

*Data availability:* The results are available in a 5 kyr (2D fields) and 100-year (scalar) output frequency at Zenodo [DOI will be added upon acceptation]. Additional 2D fields and higher output frequencies up to 1 kyr can be requested by contacting the corresponding author.

*Author contributions.* MS conducted the simulations and has written the manuscript. The set-up for the experiments was created by RW, CB and MS. CB provided model support. All authors have provided input to the manuscript and analysis of the results.

*Competing interest.* The authors declare that they have no conflict of interest.

*Acknowledgements.* M.D.W. Scherrenberg is supported by the Netherlands Earth System Science Centre (NESSC), which is financially supported by the Ministry of Education, Culture and Science (OCW) on grant no. 024.002.001. C.J. Berends is funded by PROTECT. This project has received funding from the European Union's Horizon 2020 research and innovation programme under grant agreement No 869304, PROTECT contribution number [will be assigned upon publication]. The Dutch Research Council (NWO) Exact and Natural Sciences supported the supercomputer facilities for the Dutch National Supercomputer Snellius. We would like to acknowledge the support of SurfSara Computing and Networking Services.

## Appendix A: Basal sliding, friction and hydrology

In this study, we have used a Budd-type sliding law to simulate the sliding at the base of the ice sheets. To obtain the friction under the base of the ice sheet, we applied a parameterisation based on Martin et al. (2011). Table 2 list the units and corresponding values of constants named here.

   The pore water pressure ($\psi$) is parameterized based on Martin et al. (2011). The pore water pressure scaling factor ($\lambda$) determines the saturation of the base and depends on the local ($x, y$ indexes) bedrock height ($b$) and concurrent sea level

($SL$):

$$\lambda = 1 - \frac{b(x,y) - SL - b_{\psi,min}}{b_{\psi,max} - b_{\psi,min}}. \tag{1}$$

The pore water pressure scaling factor is limited between 1 (fully saturated) and 0 (fully unsaturated). Here, $b_{\psi,max}$ is the bedrock elevation with respect to sea level at which beds become fully unsaturated (1000 m), while $b_{\psi,max}$ is the elevation when beds become fully saturated (0 m). This pore water pressure scaling factor is used in combination with the overburden

pressure to calculate the pore water pressure ($\psi$):

$$\psi = P_w \, \rho \, g \, H(x,y) \, \lambda. \tag{2}$$

Here, $H$ represents the thickness of the ice. $P_w$ is the pore water pressure scaling fraction. Here we have used 0.99, which yields a good LGM ice sheet volume and extent in combination with the prescribed till friction map. To determine the till friction angle ($\phi$), we use different data-sets for North America and Eurasia. For North America we use a geology reconstruction from

Gowan et al. (2019), which has full coverage of our ice domain and indicates were bedrock could more easily erode. The map from Gowan et al. (2019) has been specifically created for ice sheet modelling. For Eurasia we have created a till friction angle map based on sediment thicknesses from Laske and Masters (1997), where till friction angles are 10 degrees when sediment thicknesses exceed 100 meters, while for other regions a value of 30 degrees is used. In the High Friction simulations, till friction angles are 30 degrees in North America and Eurasia, while the Low Friction simulations has till friction angles of 10

degrees. For Greenland, which is not the focus of our experiments, we always use the same till friction angle map of 30 degrees for land at present-day and 10 degrees for ocean. The pore water pressure and till friction angle can then be used to calculate the till yield stress ($\tau$):

$$\tau(x,y) = \tan\left(\frac{pi}{180}\right) \phi \, (\rho \, g \, H(x,y) - \psi) \tag{3}$$

To calculate the basal friction coefficient ($\beta$), we use the Budd type sliding law (Bueler and van Pelt, 2015):


$$\beta(x,y) = \tau(x,y) \frac{u^{(q-1)}}{u_0^q} \tag{4}$$

Here, $u$ represents the ice velocity. The sliding term, is then used to calculate the basal sliding. The basal friction coefficient is multiplied by the grounded fraction, which is calculated based on Leguy et al. (2021). Therefore, regions that are fully floating (grounded fraction of 0) receive a negligible friction, the grounding line (grounded fraction between 0 and 1) receive

reduced friction, and fully grounded ice receive the full friction (grounded fraction of 1). For the Rough Water simulation, the
basal friction coefficient is always multiplied by 1 instead and therefore the friction treats all ice as if it were grounded.

**Table 2:** Constants describing till friction angle and basal hydrology

| Symbol | Description | Units | Value |
|--------|-------------|-------|-------|
| $b_{\psi,min}$ | Elevation with fully saturated beds | m | 0 |
| $b_{\psi,max}$ | Elevation with fully unsaturated beds | m | 1000 |
| $\rho$ | Density of ice | kg/m$^3$ | 910 |
| $g$ | Gravitational acceleration | m/s$^2$ | 9.81 |
| $P_w$ | Pore water pressure scaling | | 0.99 |
| $q$ | Exponent in Budd type sliding law | | 0.3 |
| $u_0$ | Threshold velocity in Budd type sliding law | m/yr | 100 |

## Appendix B: Climate time-slice interpolation

To provide the ice-sheet model with transiently changing forcing using minimal computational resources, we interpolate
between pre-calculated LGM and PI climate time-slices. To interpolate the time-slices we have used a matrix method. Our
approach is based on Berends et al. (2018) and Scherrenberg et al. (2023) and uses different methods for temperature and
precipitation.

To calculate the temperature forcing, we use a linear interpolation:

$$T(x, y, mnth) = w_T(x, y)\, T_{PI}(x, y, mnth) + \left(1 - w_T(x, y)\right) T_{LGM}(x, y, mnth). \tag{5}$$

Here, $T$ is the monthly ($mnth$) temperature forcing in the ice sheet model. $T_{PI}$ and $T_{LGM}$ are the climate model temperatures
for PI and the LGM respectively. $w_T$ is the interpolation weight and depends on two processes: the external forcing ($w_e$) and
an albedo feedback ($w_a$). $w_T$ is capped between -0.25 and 1.25 to prevent too much extrapolation. For North America and
Eurasia, $w_T$ is calculated as following (see Berends et al. 2018):

$$w_T(x, y) = \frac{w_e(x,y) + w_a(x,y)}{2}\ . \tag{6}$$

In Greenland the albedo changes almost exclusively due to the change in ice-sheet extent. Our model does not include sea
ice and the Greenland domain does not contain extensive tundra areas. Therefore, we apply a smaller contribution from
albedo:

$$w_T(x, y) = \frac{3\, w_e(x,y) + w_a(x,y)}{4}\ . \tag{7}$$

To calculate $w_e$ we combine the effect of $CO_2$ (ppm; Bereiter et al., 2015) and insolation at 65°N ($Q_{65°N}$; W/m²; Laskar et al., 2004):

$$w_e = \frac{C - C_{LGM}}{C_{PI} - C_{LGM}} + \frac{Q_{65°N} - 440\ W/m^2}{70\ W/m^2}. \tag{8}$$

In this equation, $C_{PI}$ and $C_{LGM}$ are 190 and 280 ppm respectively. By including the summer (June, July, August) insolation at 65°N ($Q_{65°N}$), the climate can become colder or warmer even if the $CO_2$ concentration is constant (see Fig. 2). We obtained the ratio between $CO_2$ and insolation by first conducting a preliminary simulation based on the method by de Boer et al. (2013) and Berends et al. (2021), where forcing was modified to obtain a match with benthic $\delta^{18}O$ from Ahn et al., (2017). This essentially reproduces the forcing needed to match the benthic $\delta^{18}O$ record. We then fitted $CO_2$ and summer insolation to this forcing to obtain Eq 8.

To calculate $w_a$, which represents an albedo feedback, we calculate the annual absorbed insolation by the surface. The absorbed insolation ($I$) depends on the monthly internally calculated surface albedo ($\alpha_s$) and insolation at the top of the atmosphere ($Q$):

$$I(x,y) = \sum_{m=1}^{12} Q(x,y,mnth)\left(1 - \alpha_s(x,y,mnth)\right). \tag{9}$$

The albedo is calculated in the ice-sheet model using Eq. 24 (see appendix D). To calculate an interpolation weight for the absorbed insolation ($w_i$), we need to calculate reference fields for the LGM and PI. To calculate the albedo for the LGM time-slice, we use land and ice masks from the ice sheet reconstruction by Abe-Ouchi et al. (2015), as these were also used in the climate model simulations. We integrate the SMB model forward through time with a fixed climate and ice-sheet geometry until the firn layer reaches a steady state (typically after ~30 years). We can then use Eq. 9 to calculate the reference fields for absorbed insolation. These absorbed insolation fields can then be used to calculate $w_i$.

$$w_i(x,y) = (I(x,y) - I_{LGM}(x,y)) / (I_{PI}(x,y) - I_{LGM}(x,y)). \tag{10}$$

To account for both the local and domain wide change in albedo and insolation, we use the following equation based on Berends et al. (2018):

$$w_a(x,y) = \frac{w_i(x,y) + 3\ w_{i,smooth}(x,y) + 3\ w_{i,domain}(x,y)}{7}. \tag{11}$$

Here, $w_{i,domain}$ is the domain-wide averaged interpolation weight and $w_i$ is the local interpolation weight. $w_{i,smooth}$ represents the regional temperature effect and is calculated by applying a 200 km gaussian smoothing on $w_i$. Once again, we use a different method for Greenland due to the lack of tundra regions:

$$w_a(x,y) = \frac{w_{i,smooth}(x,y) + 6\ w_{i,domain}}{7}. \tag{12}$$

The interpolation weight to calculate temperature ($w_T$) can now be derived from $w_a$ and $w_e$ using Eq. 6 or 7. This interpolation weight will change depending on albedo, insolation and $CO_2$.

For precipitation, we apply a different method, as the precipitation does not change linearly when the climate cools down and topography changes. We use the following equation to interpolate the precipitation from the climate time-slices:

$$P = exp\left((1 - w_P(x,y))\log\left(P_{PI}(x,y,mnth)\right) + w_P(x,y)\log\left(P_{LGM}(x,y,mnth)\right)\right). \tag{13}$$

$w_P$ is the interpolation weight and depends on local and domain-wide topography changes. First, we compare the domain-wide topography in the model to the climate time-slices using the following equation:

$$w_{s,domain} = (\sum s(x,y) - \sum s_{PI}(x,y)) / (\sum s_{LGM}(x,y) - \sum s_{PI}(x,y)). \tag{14}$$

The surface topography is represented by $s$. $w_{s,domain}$ represents the interpolation weight from the domain-wide change in topography. If a grid-cell was covered with ice during the LGM, we also interpolate with respect to local changes in topography:

$$w_{S,local}(x,y) = \frac{S(x,y) - S_{PI}(x,y)}{S_{LGM}(x,y) - S_{PI}(x,y)} \, w_s(x,y). \tag{15}$$

If a grid-cell did not have ice during the LGM, $w_{s,local}$ is equal to $w_{s,domain}$. In the last step, we multiply the local and regional precipitation effect to obtain the interpolation weight for precipitation:

$$w_P(x,y) = w_{S,local}(x,y) \, w_{S,domain}(x,y). \tag{16}$$

The resulting $w_P$ from Eq. 16 is used in Eq. 13 to calculate the interpolated precipitation forcing.

## Appendix C: Downscaling and bias correction

To account for differences between the general circulation model (GCM) simulations and observed climate (ERA40; Uppala et al., 2005), we apply a bias correction on both the LGM and PI snapshots.

To account for the temperature bias, we first have to scale the temperature to sea level using a lapse-rate correction. This is to account for differences in topography between the GCM and ERA40 data.

$$T_{obs,SL}(x,y,mnth) = T_{obs}(x,y,mnth) + s_{obs}(x,y) \lambda(x,y). \tag{17}$$

$$T_{GCM,SL}(x,y,mnth) = T_{GCM,PI}(x,y,mnth) + s_{GCM,PI}(x,y) \lambda(x,y). \tag{18}$$

Here, T is the temperature from ERA40 (obs) and the pre-industrial (PI) time-slices of the climate model (GCM). Surface height is defined as $s$. The temperature lapse rate ($\lambda$) is equal to 0.008 K/m. Once the temperature is applied to sea level (SL), we calculate the temperature difference between the climate model and observed climate:

$$T_{GCM,bias}(x,y,mnth) = T_{GCM,SL}(x,y,mnth) - T_{obs,SL}(x,y,mnth). \tag{19}$$

This bias correction is then subtracted from the PI and LGM snapshots. As a result, the PI snapshot will be the equal to ERA40, which contains some anthropogenic warming.

For precipitation ($P$), biases are applied as ratios rather than absolute differences, to ensure that the bias-corrected values are always positive. Therefore, we use the ratio between the model and observed fields instead:

$$P_{GCM,bias}(x,y,mnth) = P_{GCM}(x,y,mnth) / P_{obs}(x,y,mnth). \tag{20}$$

This ratio is used to calculate the bias corrected precipitation for PI and LGM:

$$P(x,y,mnth) = P_{GCM}(x,y,mnth) / P_{GCM,bias}(x,y,mnth). \tag{21}$$

## Appendix D: Surface mass balance model

The surface mass balance (SMB) is calculated using an insolation-temperature model; IMAU-ITM (Berends et al., 2018). To calculate the SMB, ice is added due to snow and refreezing and is removed due to melt. To calculate accumulation and ablation of ice, the model requires temperature and precipitation fields, which were obtained from downscaled and bias-corrected GCM output (see appendix B and C). To calculate the amount of snowfall, we apply a temperature-based snow-rain partitioning with respect to the melting point ($T0$) by Ohmura et al. (1999).

$$f = 0.5\left(1 - \frac{\text{atan}\frac{(T(x,y,mnth)-T_0)}{3.5}}{1.25664}\right). \tag{22}$$

The snow fraction ($f$) determines the amount of precipitation that falls as snow; the remainder falls as rain. The snow fraction is limited so it always is between 0 (100% rain) and 1 (100% snow). $x$ and $y$ indicate the horizontal grid while $m$ indicates the month. To calculate the ablation of ice, we use the parameterised scheme by Bintanja et al. (2002) that accounts for ablation from temperature and insolation:

$$M(x,y,m) = (c_1\,(T(x,y,mnth) - T_0) + c_2\left(1 - \alpha(x,y,mnth)\right)Q(x,y,mnth) - c_3). \tag{23}$$

Here, T is the 2-meter air temperature, $T_0$ is the melting temperature of ice or 273.16 K, $Q_{TOA}$ is the insolation at the top of the atmosphere (Laskar et al., 2004). The parameters for $c_1$, $c_2$ are 0.079 m/yr/K and 7.9x10$^{-4}$ m/J/yr. The parameter $c_3$ is used for tuning. Here we have tuned the model to obtain realistic LGM ice volumes, with $c_3$ values for North America (-0.16 m/yr) Eurasia (-0.24 m/yr) and Greenland (0.19 m/yr). Albedo ($\alpha$) is calculated in the ice-sheet model and is also based on Bintanja et al. (2002):

$$\alpha(x,y,mnth) = \alpha_{snow} - (\alpha_{snow} - \alpha_b)\exp^{-15\,D(x,y,mnth-1)} - 0.015\,M_{prev}(x,y).. \tag{24}$$

The $\alpha_b$ represents the albedo without any snow, with 0.5 for bare ice, 0.2 for land and 0.1 for water. The amount of melt (m) during the previous year is defined as $Melt_{prev}$. If snow is added on top, which increases the depth (m) of the firn layer ($D$), the albedo can increase until $\alpha_{snow}$, which represents the albedo of fresh snow (0.85). Therefore, the albedo in the model varies between the background and snow albedo. The depth of the firn layer is calculated using the amount of snow that is added on top without melting.

Some of the melt and rainfall can refreeze in the model. Here we use the approach by Huybrechts and de Wolde (1999), using the total amount (m/yr) of liquid water (l), superimposed water (s) and precipitation (P).

$$s(x,y,mnth) = \max\{0, 0.012\left(T_0 - T(x,y,mnth)\right)\}. \tag{25}$$

$$l(x,y,mnth) = R(x,y,mnth) + M(x,y,mnth). \tag{26}$$

$$r(x,y,mnth) = \min\{\min\{S(x,y,mnth), L(x,y,mnth)\}, P(x,y,mnth)\}. \tag{27}$$

By combining the snowfall, refreezing and melt the SMB can be calculated:

$$SMB(x,y,mnth) = S(x,y,mnth) + r(x,y,mnth) - M(x,y,mnth).\quad. \tag{28}$$

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

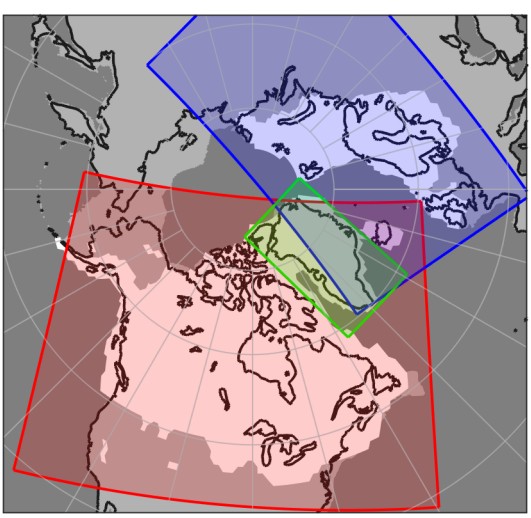

**Figure 1.** The extent of the North American (red), Greenland (green) and Eurasian (blue) domains. The present-day coastline is shown (black lines), as well as the LGM land and ocean (shown in grey). The extent of the LGM ice sheets in Abe-Ouchi et al. (2015) is shown in white.


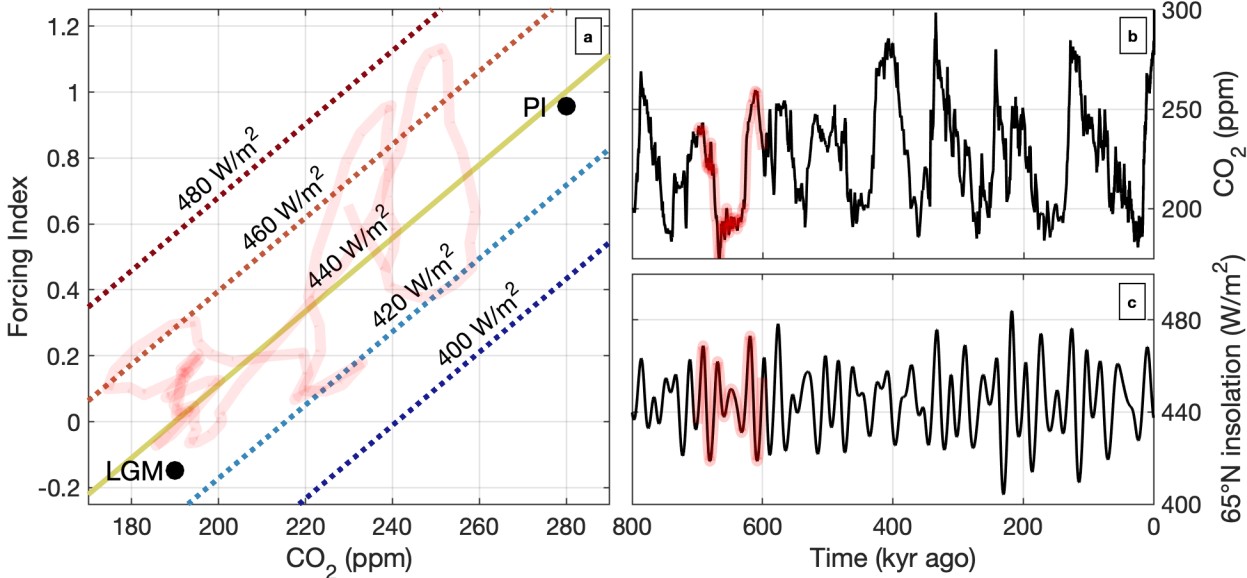

**Figure 2.** The forcing index (a), which combined with an albedo feedback, drives temperature changes in the ice-sheet model. The forcing index depends on the prescribed $CO_2$ (b; Bereiter et al., 2015) and insolation (c; Laskar et al., 2004). The pathway of the forcing-index for a 100-kyr period is shown in red. A forcing index of 0 (1) represents LGM (PI) temperature contribution from external forcing.

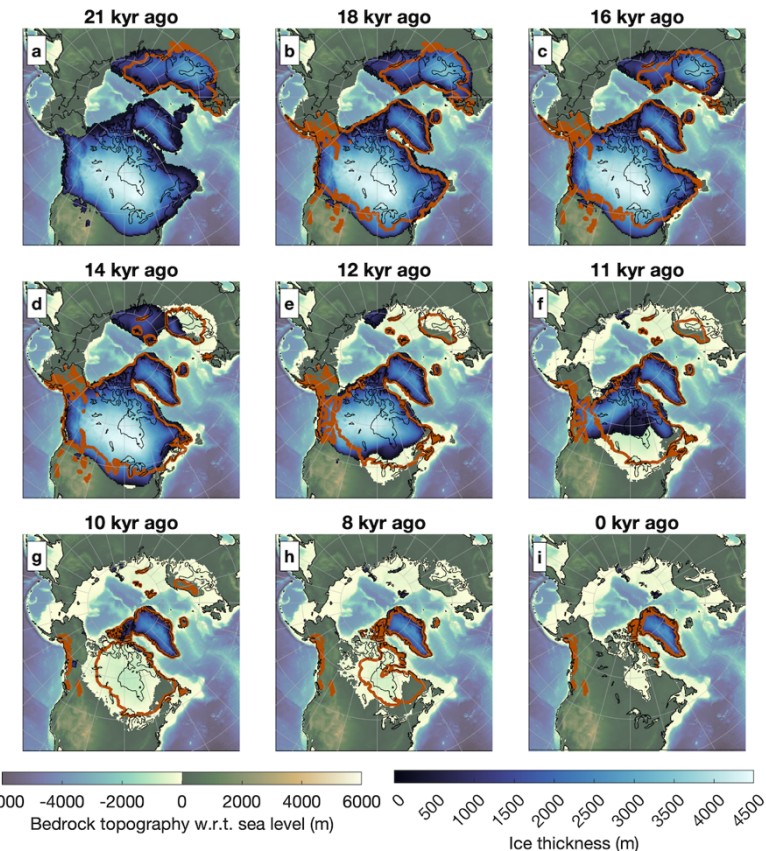

**Figure 3.** Ice thickness and bedrock topography of the Baseline simulation during the last deglaciation. The North American ice sheet from Dalton et al. (2020) and the Eurasian ice sheet reconstruction from Hughes et al. (2015) are shown as orange contours, while the present-day coastline is shown as black contours.

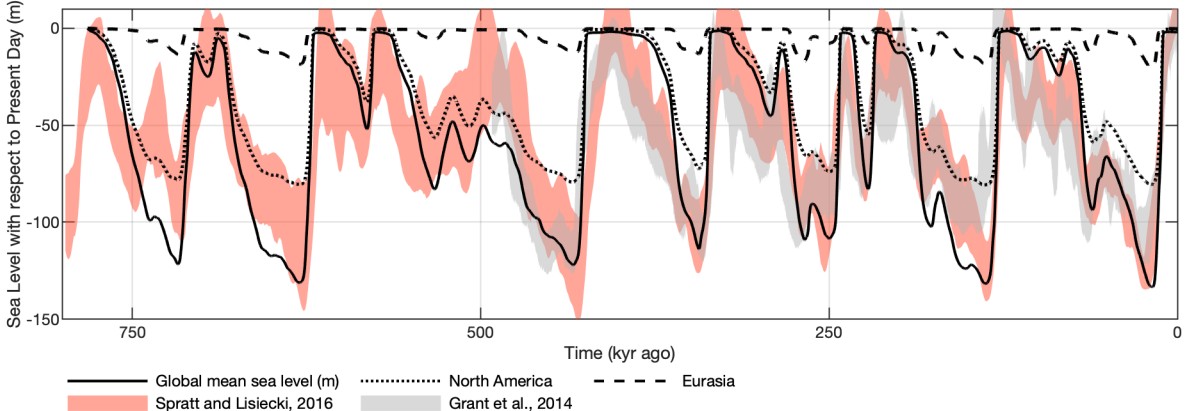

**Figure 4.** Simulated global mean sea level of North American (dotted line) and Eurasian (dashed line) of the Baseline experiment. Global mean sea level change (solid line) is the volume changed from the Northern Hemisphere ice sheet and an additional 30% to represent other ice sheets (e.g., Patagonia, Himalaya, Antarctica). These are compared to Grant et al. (2014) and Spratt and Lisiecki (2016).

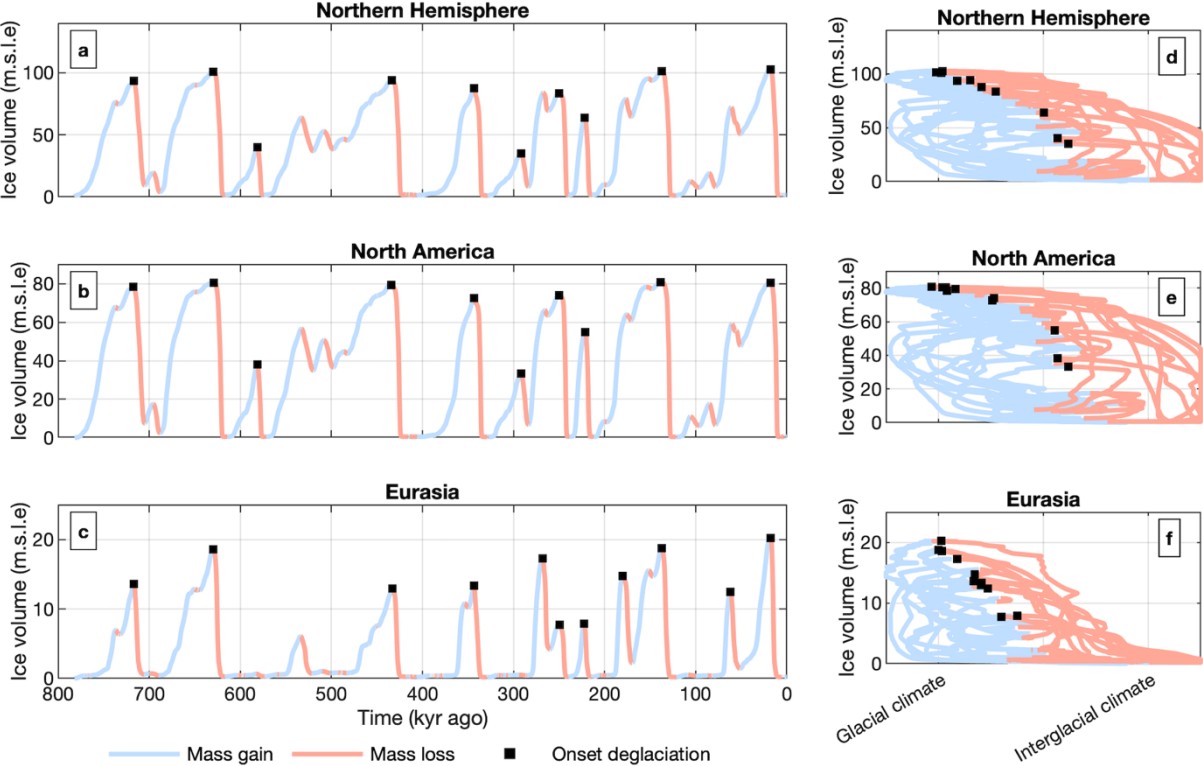

**Figure 5.** Time series of the simulated ice volume in the Baseline experiment are shown in panels a-c. Panels d-f show the ice volume with the corresponding climate forcing (glacial to interglacial) based on the external forcing (prescribed $CO_2$ and insolation forcing). Panel a and c show the ice volume of Eurasia, Greenland and North-America combined. Colours indicate net melt (red) net accumulation (blue) and onset of deglaciation (black squares). The onset of deglaciation is only placed if the volume is at least 30% at the onset, and lower than 30% by the end of the termination compared to the maximum of the simulation.

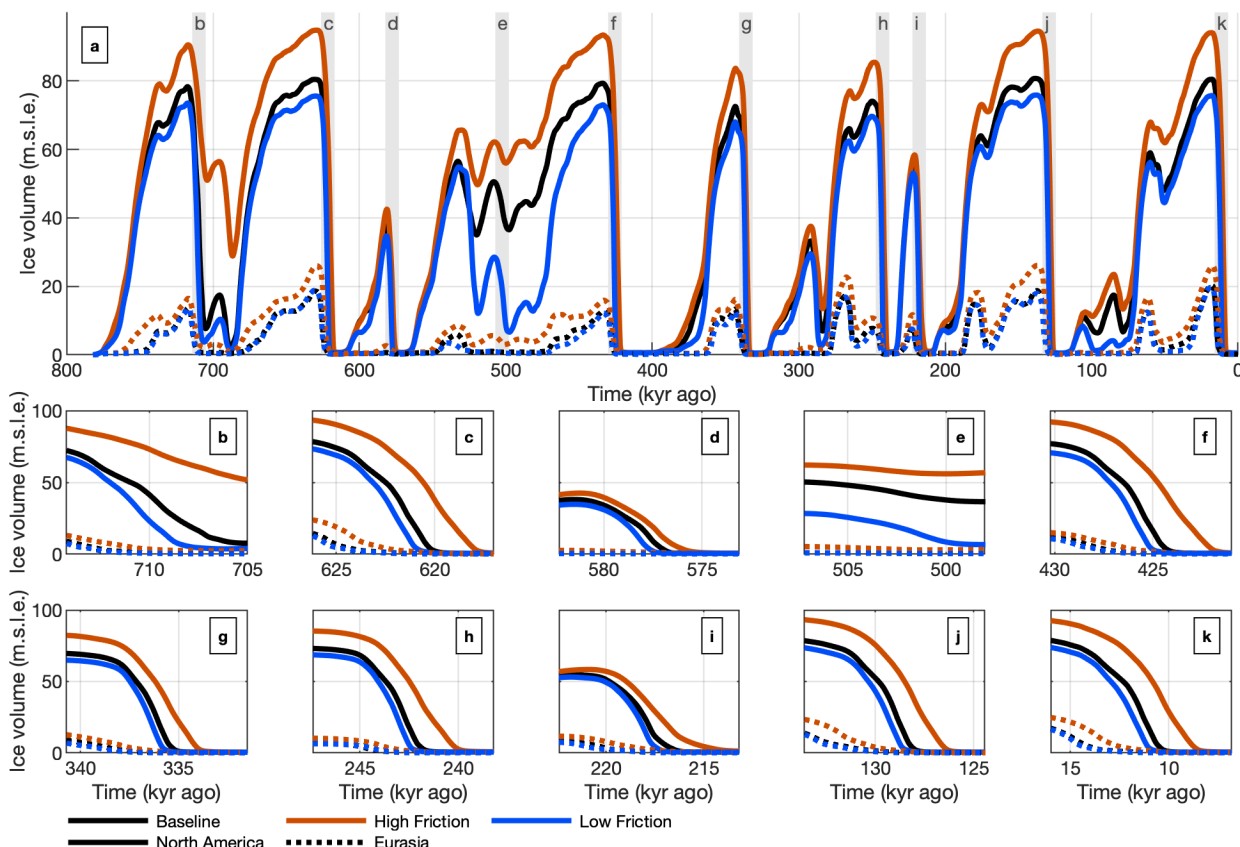

**Figure 6.** Ice volume (m.s.l.e) time-series of the Baseline, High Friction and Low Friction are shown in panel a. Grey regions in panel a indicate the regions of the zoomed in time-series shown in panels b-k.

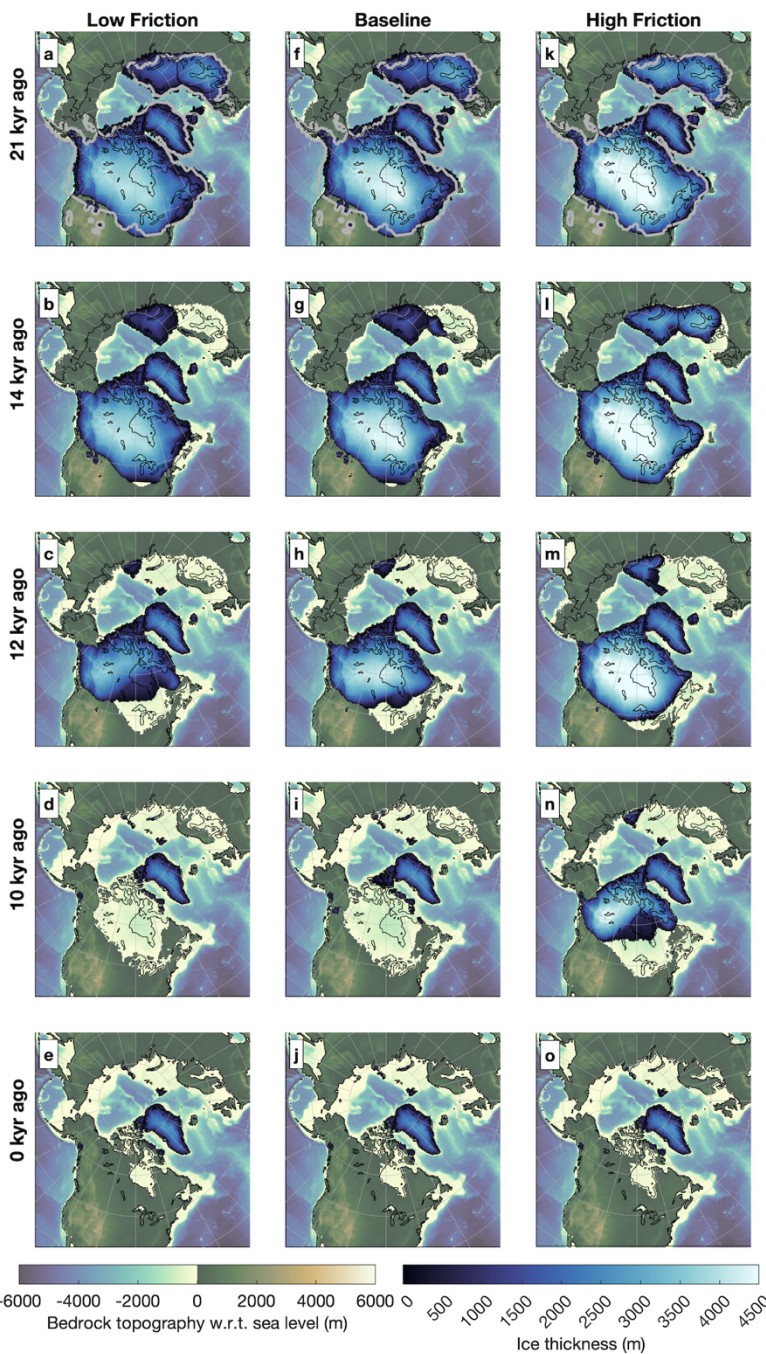

**Figure 7.** Ice thickness and bedrock topography of the Low Friction (a-e), Baseline (f-i) and High Friction (k-o) experiments. Time-slices for 21 thousand years ago (kyr; a,f,k), 12 kyr ago (c-m) and 10 kyr (d-n) are shown. The LGM extent from Abe-Ouchi et al. (2015) is shown as grey contours, whereas black contours indicate the present-day coastlines.

990

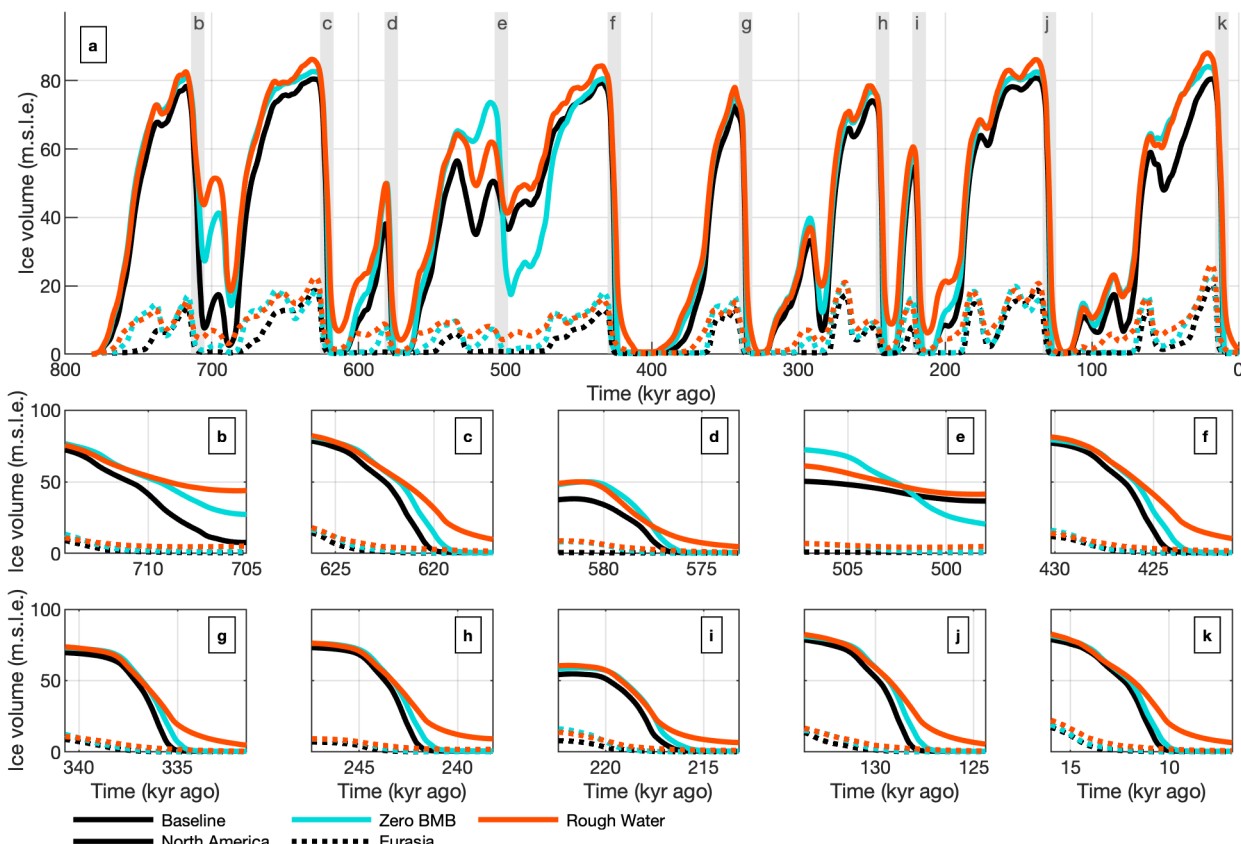

**Figure 8.** Time-series of the North American and Eurasian ice sheets during various deglacial periods. The full 800 kyr time-series is shown in panel a. The grey patches in panel a correspond to the time-series shown in panels b-k.

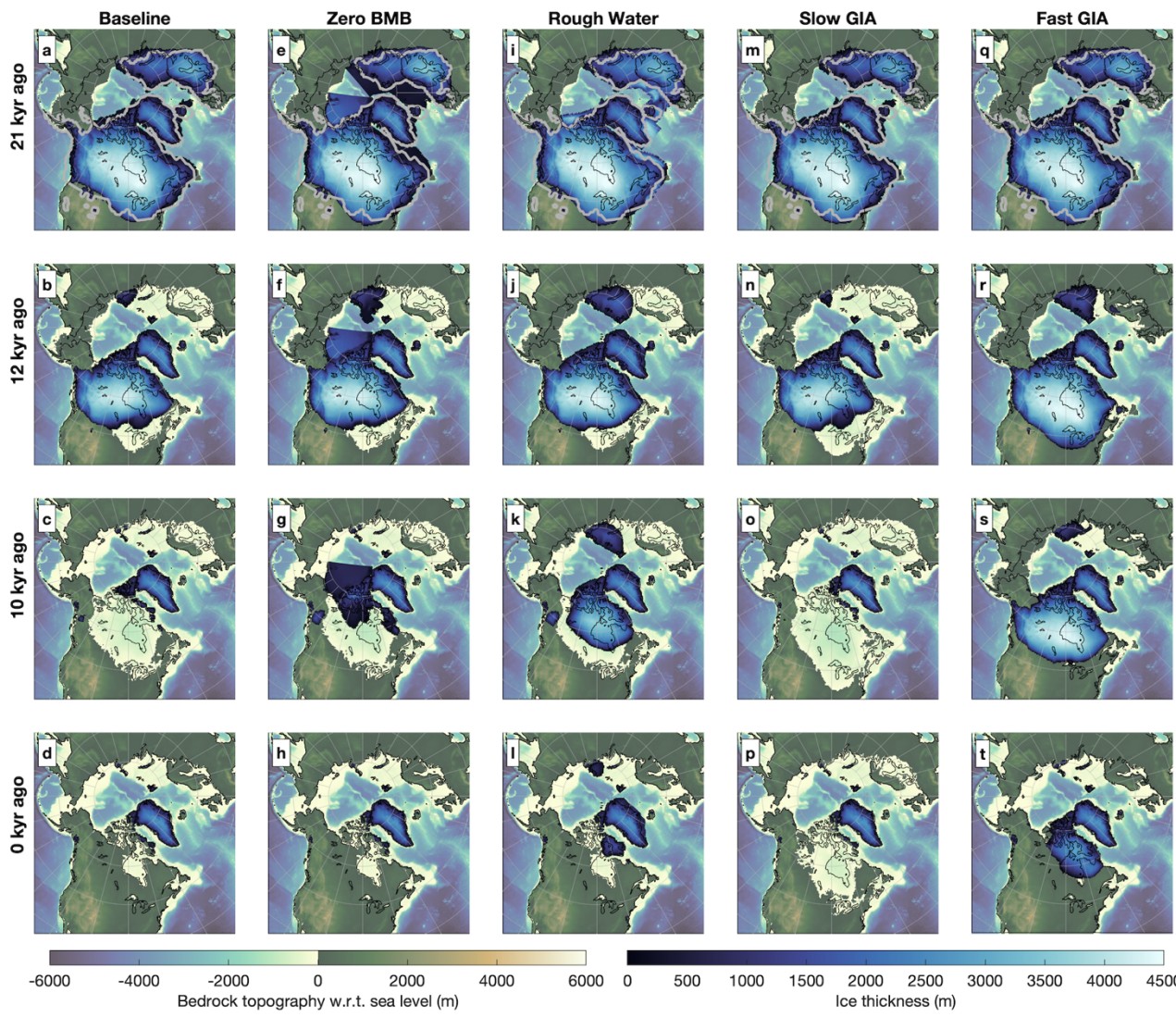

**Figure 9.** Bedrock topography and ice thickness for the Baseline, Rough Water, Slow GIA and Fast GIA simulations (columns) for 21 ka, 11 ka, 9 ka and 0 ka (rows). The reconstruction from Abe-Ouchi et al. (2015) is shown in grey and the present-day coastline is shown in black.

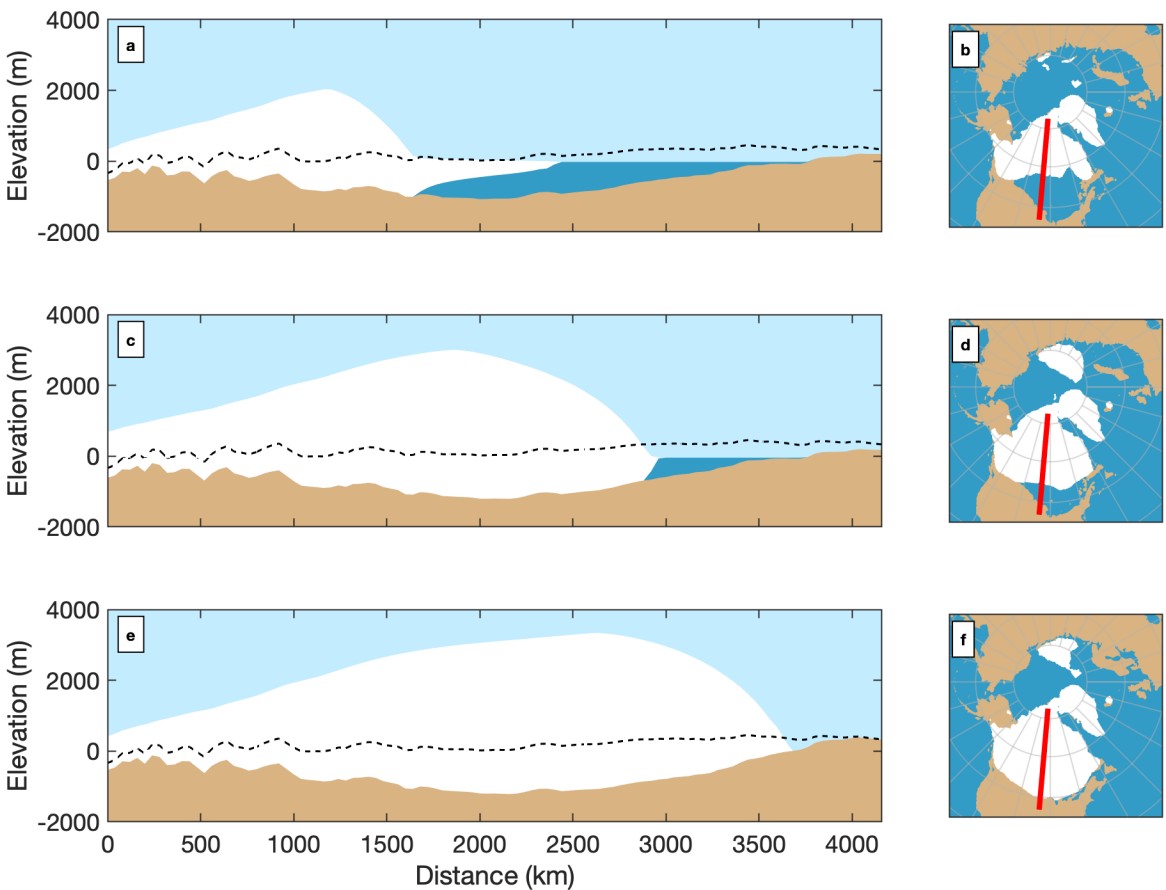

1005

**Figure 10.** Transects (a,c,e) of the Baseline (a,b) the Rough Water (c,d) and Fast GIA simulations (e,f) at 11 kyr ago. Present-day bedrock is shown as a dashed line in a,c and e. The 0 km distance represents the Northern-most point of the transect, which is shown in panels b,d and f.

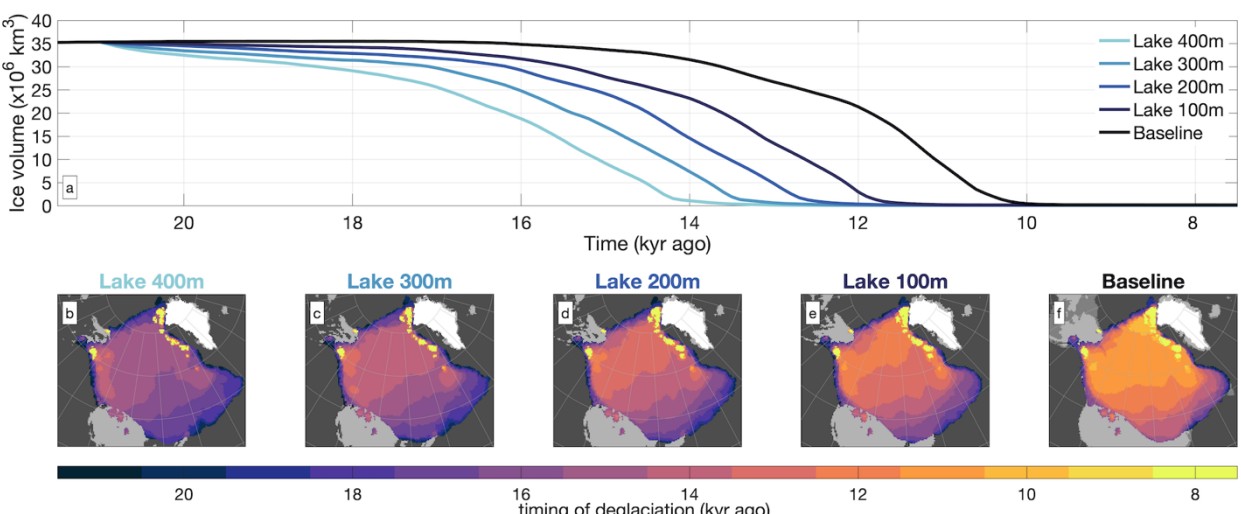

**Figure 11.** Ice volume time-series of varying lake levels compared to the Baseline. Panels b-f shows the timing of deglaciation. The background colours indicate, from dark to light, ocean during LGM and PI (dark-grey), land during LGM, but not at PI (grey), land in both LGM and PI (light-grey), and the present-day ice coverage (white).

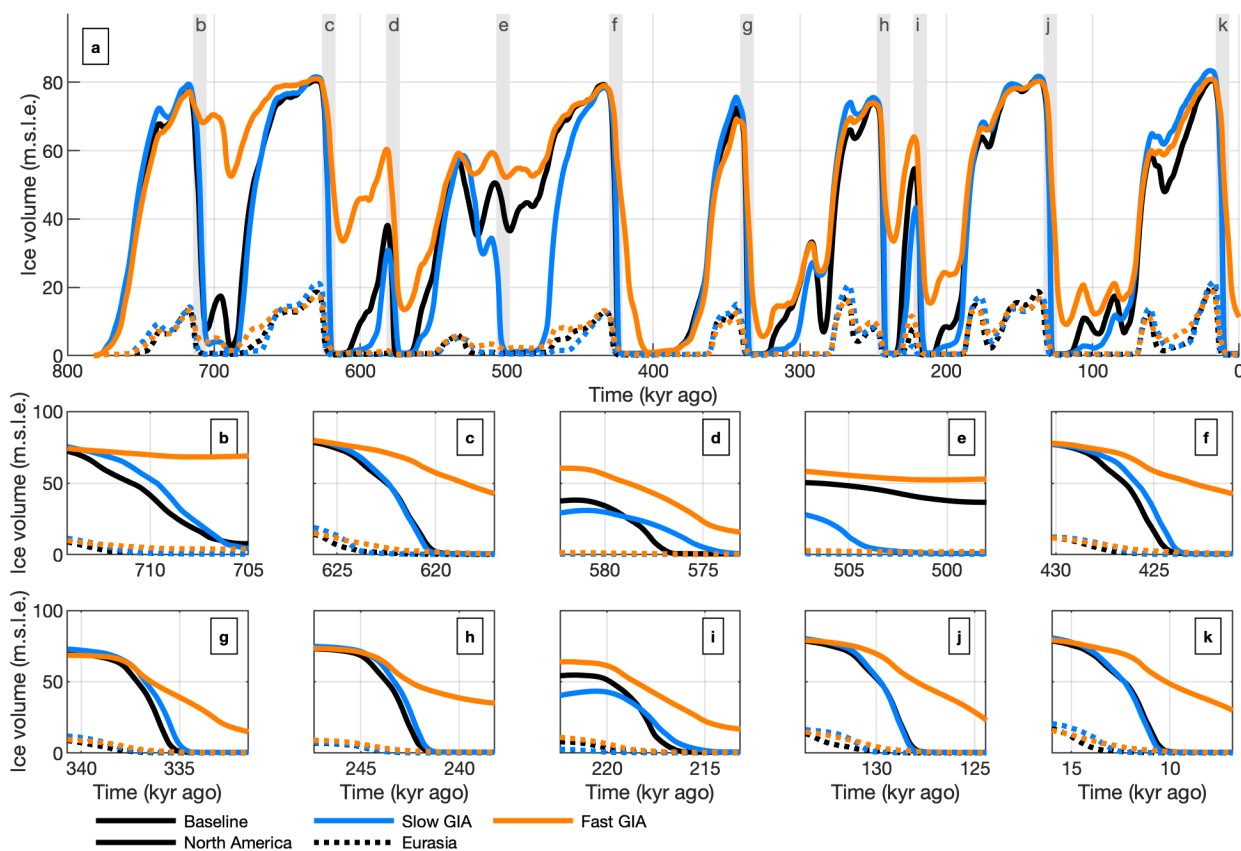

**Figure 12.** Time-series of ice volume with different GIA relaxation times. The time-series of the past 800 kyr is shown in panel a. Panels b-k show one termination each, which are indicated by the grey patches in panel a.