# Peer review of "Late Pleistocene glacial terminations accelerated by proglacial lakes"

_Climate of the Past, 2023_

## Author Comment (AC1)

Response to the review by Fuyuki Saito

We thank the reviewer for constructive and helpful commentaries. We would hereby like to address the concerns raised. Reviewer comments are shown in bold and our responses are shown in regular font type.

**Description of the ice-sheet model: Section 2.1 describes the ice-sheet model briefly, but I feel it too short. A complete repeat of the previous papers are not necessary, but at least the description of the methods relating to the sensitivity experiments of the present paper should be included in detail.**

We will improve the model description section based on the suggestions listed below (treatment of grounding line migration, modelling of proglacial lakes, and basal sliding).

Additionally, we will add an appendix which describes the till friction angle, basal hydrology and resulting basal friction.

**(i) how to deal the grounding line migration in the model. it is all right without any special treatment, but mention explicitly.**

We use a sub-grid friction scaling scheme to achieve good grounding-line dynamics at relatively coarse resolutions, similar to e.g. CISM (Leguy et al., 2014; The Cryosphere) and PISM (Feldmann et al., 2014; Journal of Glaciology). The IMAU-ICE model description paper (Berends et al., 2022; Geoscientific Model Development) showed that this enables IMAU-ICE to resolve the migrating grounding line to within a single grid cell. We will add several sentences explaining this in more detail.

**(ii) how to deal the formation of proglacial lakes. I suppose that the depth of lake is the difference between the background sea level and the bedrock.**

This is correct. We will explain these in more detail in the method section of the revised manuscript, and justify our choice for this simplification.

**(iii) explicit equation of basal sliding, and the difference in how to apply between base-line and rough-water experiments.**

The equations governing the basal roughness, basal hydrology, and the sliding law, will be added in an appendix. We will also include a more thorough description of the Rough Water simulation, in which the basal friction is treated the same regardless if the ice is floating or not.

**Other points:**

**Abstract: There are no description of the method used in the present paper, even no explanation that the discussion is based on a series of numerical simulation. This must be clarified.**

This is indeed an oversight in the abstract. We will add a few sentences describing the methods in the abstract. These include 1) explicitly mention we have conducted a numerical modelling study and 2) that we varied the representation of certain processes in this model to determine the effects of GIA, ice-dynamics and basal melt.

**BMB experiments: Typical mass balance terms, both surface and base, over ice shelves in the baseline experiment should be mentioned, which will help to get typical ratio of surface/basal mass balance terms.**

We will add a figure to the paper comparing BMB, SMB and ice volume to the supplementary information.

**Figure 2. Plotting two points corresponds to PI and LGM may help.**

We will add a point for LGM and PI. We have tuned the insolation to be able to simulate glacial cycle periodicity and obtain a reasonable LGM volume and melt at present-day. Though, since PI and LGM have different insolation at 65N, at 280 ppm $CO_2$ and PI insolation will be close to, but not exactly match, present-day conditions.

**Figure 3. Better to mention in the caption that this is the result of baseline experiment.**

We will add "Baseline" to the caption to show that this is indeed the Baseline experiment.

**Figure 4 and main text. Please note the definition of the onsets of (de)glaciation in this paper. Why some points (e.g., 240ka in the total and Eurasia, 290ka in North America) are not detected as the onsets?**

The onset and termination points were based on a few thresholds. First of all, the ice sheet needed to be big enough and melt for a long enough duration. However, we have decided to change these thresholds and rework the figure.

The following changes will be made to improve the figure.

1) The red "onset of deglaciation" points are now located at the right location (so only at glacial maxima).

   An "onset of deglaciation" point is now added when the ice sheet melts at a large enough volume (at least 20% of the modelled Late Pleistocene

maximum) and has melted enough ice (less than 20% of the maximum Late Pleistocene volume remaining). These thresholds will be added to the caption.

2) The blue "onset of glaciation" points have been removed as they added little to the overall story while making the figure more complicated.

3) "External forcing index" is replaced by "glacial" and "interglacial" climate to make the figure easier to understand.

4) Figure 7 (a similar figure) has been removed and replaced by 2D ice thickness maps.

We have added the reworked figure below which shows all the changes mentioned above:

[Figure]

**Appendix Eq.(1) Please clarify that f_{snow} is no more than 1.  I calculated by hand, and observed that f_snow become 1 when T - T0 is around -10K, and exceeds 1 when -11K.  I am not sure my rough computation is correct, but please clarify the actual computation of f_{snow} (Of course it is all right if T - T0 is always more than -10K).**

f_snow in the model is limited between 0 and 1, though this was not stated in the manuscript. Therefore, f_snow will not exceed 1 even when temperatures are well below 262K. We will add a sentence clarifying that f_snow is limited between 0 and 1.

**Melt_{prev}.  Again, it is hard to check whether the albedo is between 0 and 1 by this equation.**

**Eq (4). Need units of the coefficient 0.012.**

**Eq (21).  Need unit(scale) of P_{PI,corr} to compute log.  Or, is the first term e means exp() function, not e times ()?**

We will add units to these equations. For equation 21, this should have been exp, and we will change it accordingly. A similar change will be made in equation 3.

---

## Author Comment (AC2)

Response to the editor comments by Lev Tarasov,

We thank the editor for constructive and helpful commentary and would hereby like to address some of the issues they raised.

**As you draft your response to reviewers, I have identified 4 other major issues that need to be addressed.**

1) **Experimental design.**

**The current design leaves too many uncertainties unaddressed in part stemming from model limitations somewhat buried or not even described in your much too brief model description (as already raised by one of your reviewers).**

The model description will be expanded. There are three processes that will be explained in more detail:

1) Treatment of proglacial lakes. We use the difference between bedrock and sea level to determine the location of lakes. We also acknowledge that as a limitation.
2) Basal hydrology was not mentioned in the method, and this should have been there from the start. An appendix will be added that explains the way basal hydrology, basal roughness and sliding law are implemented in our model and presents the governing equations.
3) The treatment of grounding line migration. Here we used a sub-grid friction scaling scheme. This process is important to all our experiments, so we will explain it in mfore detail.

Additionally, each simulation has slightly different ice volumes at glacial maxima, which has an effect on the deglaciation. This effect is small, but difficult to completely disentangle, and will be acknowledged.

We have also conducted High and Low Friction cases for Fast GIA and Rough Water to show that our main conclusions are consistent regardless of basal friction.

We will furthermore mention this limitation in more detail, as well as indicate that there are more sophisticated methods to deal with basal friction (e.g., sediment mask / models) and basal hydrology.

**1a)**

**As far as I can tell, you do not use a soft/hard bed mask for Eurasia, even though it has long been known that this can have a large impact on ice sheet response and geometry (eg cf Tarasov and Peltier, 2004). So a significant**

**fraction of your sensitivity to basal drag changes from pro-glacial lakes could be due to this easy to address design limitation as regions that are soft-bedded will generally have less basal drag (when warm-based).**

For basal friction we use a parameterization from Martin et al. (2011). However, we do not use a sediment map to calculate basal friction.

Therefore, we will add two additional experiment to investigate the importance of basal friction on the deglaciation.

We perform two new experiments by multiplying the till friction angle with 1.5 (High Friction) and 0.5 (Low Friction). A new paragraph will be written to show the results of these two additional experiments. These include showing a time-series as well as 2D ice thickness / bedrock topography maps of the last deglaciation.

The same friction experiments will also be applied for Rough Water and Fast GIA, adding an additional four experiments. Though the results of these four experiments will be added to the supplementary information instead.

We found that the basal friction has a substantial effect on ice volume as well as the deglaciation and we will mention that this shows the importance of basal friction / hydrology on modelling glacial cycles.

To briefly summarise the results of the friction experiments:

With increasing friction, ice volume increases (at LGM a 14% increase for high friction and 16% decrease for low friction). The differences in extent at glacial maxima is similar regardless of the basal friction use here (e.g., at LGM we found <5% difference). An increase in friction will cause a slower deglaciation.

However, basal friction remains a substantial limitation to these experiments, and will be addressed in the discussion section.

**1b)**

**I suspect you have no basal hydrology (again model description is way to brief on relevant aspects of the model configuration) and**

**I also suspect you are simply taking the adjacent water depth to compute effective pressure for the Budd sliding law. On a 40km grid, that choice is hard to defend. At the very least, try an easy to implement leaky bucket approach ( eg as in PISM or https://tc.copernicus.org/preprints/tc-2022-226/) for comparison.**

Our basal hydrology parameterisation is based on Martin et al., 2011 (The Cryosphere), and has been used in PISM. Here, the pore water pressure is calculated based on the ice thickness, bedrock topography and sea level.

Currently, the basal hydrology has not yet been explained in the model description, and will be added when revising the manuscript. A new appendix section will be added to describe the basal hydrology, till friction angle, resulting basal friction and sliding law.

**1c) The assumption that lake depth is given by bed depth below contemporaneous sealevel is another major limitation, eg long known in the geological community (or cf Tarasov and Peltier 2006 from a modelling perspective) from the critical dependence on controlling sill elevation for eg Laurentide Lake Agassiz. Surface drainage solvers have been in some published models for almost two decades. I can understand adding one is non-trivial, but at the very least more thought needs to be put into addressing the some of the uncertainties arising from this model limitation.**

To address these issues, we propose an alternative experiment to quantify the effect of this simplification. We select a large region in North America (south of the Hudson Bay to southern margin of the domain) where the water level of potential proglacial lakes is set to 50m above present-day sea level.

The results are shown below compared to the Baseline, Zero BMB and Rough Water simulations. The Lake 50m experiment loses more mass during interstadial periods, but is very similar compared to the Baseline during the deglaciation, advancing the deglaciation by at most a few centuries. This effect is small compared to the Faster GIA and Rough Water experiments.

[Figure]

**2) Scientific precision/accuracy/transparency**

**Eg:**

**"We find that the modelled sea level matches the reconstructions well"**

**Your model fails to capture any signal of last glacial inception, so blanket use of "matches the reconstructions well" is inaccurate.**

We will weaken this too strong statement. Glacial inception is not the key focus in these simulations, and could be more addressed in more detail.

**No where do you show your LGM ice extents (only 11 ka), instead your only LGM ice extent map shows a cited map from Abe-Ouchi et al (2015). This may suggest to the reader that you do not want them to see what our LGM ice extent looks like.**

We will add maps of the LGM ice sheet – as well as other time-slices – to the manuscript.

Our ice sheet extent at LGM matches reconstructions reasonably well, compared to other freely-evolving paleo-ice-sheet models (e.g., Berends et al., 2018. Willeit et al., 2019). Though, we lack substantial ice coverage in Scotland and Southern Alaska.

This mismatch is not currently described in our results, but it will be added to section 3.1.

**Or**

**"our simulations tend to have slightly too long interglacial periods compared to reconstructions" From what I can read off of your plot in figure 3, your last interglacial is @ 50kyr too long, that is not "slightly".**

Interglacial periods are too long in our simulations and we will emphasise this more.

**Adequate referencing of past relevant litterature, eg role of GIA in deglaciation of ice sheets (eg Tarasov and Peltier, Ann. Glac., vol. 25, 58-65, 1997), impact of pro-glacial lakes on ice extent (eg Cutler et al, 2001, Geology), or relative sensitivity of Eurasian versus North American ice sheets to orbital/CO2 forcing (eg Tarasov and Peltier, JGR 1997).**

We have added these, and several other relevant references for these topics.

**4) Claims in conclusions that havxe not been shown, eg: "3.4 Glacial isostatic adjustment … The North American ice sheet may not even fully deglaciate during some interglacial periods. This is because proglacial lakes are not created when the bedrock uplift is too fast". This is not shown in your results (which would require a further sensitivity experiment with lakes turned off for soft/hard GIA experiments).**

We will add 2D maps with time-slices from the last deglaciation comparing the Fast GIA, Slow GIA and Baseline simulations. These figures show that proglacial lakes become smaller with increasing uplift rates. The Fast GIA simulation will show that the proglacial lakes are much smaller and mostly absent compared to the Baseline simulation.

Additionally, we have conducted the Fast GIA, Rough Water and Baseline simulations with a 50% higher and 50% lower till friction angle. While the till friction angle does have a substantial effect on the timing of deglaciation, it does not change the main conclusion of the paper: The Fast GIA has a slower deglaciation compared to the Rough Water experiments, and the Baseline has the fastest deglaciation regardless of till friction angle.

Results of these additional friction experiments using the Fast GIA and Rough Water will be added to the supplementary information. Similar friction experiments with the Baseline simulations will be discussed in a new paragraph in the Results section.

**Furthermore, ice volume is a very limited metric. Lake calving will more directly affect ice extent than ice volume (though the two are obviously related). Without relevant map-plots (at least in a supplement), the reader is unable to adequately evaluate the extent to which you results match your claims.**

Indeed, we will add 2D ice thickness maps to the manuscript showing the last deglaciation of the Rough Water, Baseline, Fast GIA and Slow GIA, as well as the newly created Friction experiments. These maps show the Northern Hemisphere ice sheets at different time-periods (e.g., last glacial maximum, 11 ka, 9ka, present-day). Additionally, we will add time-series of ice volume compared to extent to the supplementary information.

The following figure is an example of what these maps will look like. (In this case for the new friction experiments.)

---

## Author Comment (AC3)

Response to the review by Niall Gandy

We thank the reviewer for their helpful comments, and would hereby like to address the concerns they raised. Reviewer comments are shown in bold and our responses in regular font type.

**Main points**

**Experimental design: I appreciate the novel approach of the experimental design, where you have made certain adjustments to better explore the ice sheet behaviour. However, the detailed justification for this is not explained sufficiently in the text, and I am left not fully convinced that your experiments explore the behaviour you intend. For example, the "Rough Water" experiment is designed to show the effect of negligible friction beneath ice shelves, but various feedbacks (surface profile, buttressing effects, a different GIA response to more grounded ice, ect) could be confusing the results. The experimental ethos could be explained in more detail.**

We hope to address these concerns with the following changes:

1) We will expand the model description in section 2.1. The way proglacial lakes are included in the model will be explained in more detail. The sub-grid friction scaling scheme, which we use to model the decrease in friction at the grounding line and floating ice, will be explained in more detail. The equations governing the basal roughness, basal hydrology, and the sliding law, will be added in an appendix.

2) We will include a more thorough description of the Rough Water simulation, in which the basal friction is treated the same regardless if the ice is floating or not.

3) We have conducted a few additional sensitivity experiments. We have conducted the Baseline simulation with a respective 50% increase and decrease in the till friction angle. We have conducted similar variations of the Fast GIA and the Rough Water experiments.

   The till friction angle has a strong effect on the deglaciation, with lower friction resulting in a faster deglaciation. We also found that our main conclusions are consistent under the different till friction angle: Fast GIA has the slowest melt compared to the Rough Water in regardless of the till friction

angle. Though the length of the deglaciation and the ice volume remaining during interglacial periods does increase with higher friction values.

4) We have added 2D maps of the modelled ice-sheet geometry, including the proglacial lakes, during the last deglaciation for the different experiments, to more clearly show the differences between the experiments.

**You could also undertake some offline ice shelf mass budgeting (at each timestep what is the flow over the grounding line, what is lost to surface melt, sub shelf melt, and calving) to disentangle the behaviour.**

We will add a figure showing integrated SMB, BMB and calving flux to the supplementary information.

**Specific points**

**18: linger > remain?**

This word indeed fits better and will be changed in the manuscript.

**67: I think it is common to conflate susceptibility to MISI/PLISI and sub-shelf melting/calving. Could you clarify the mechanistic difference for the reader before this sentence?**

We will add a few sentences around line 50 explaining sub-shelf melting and calving. This is before MISI is explained, and should help to prevent confusion between PLISI and sub-shelf melting/calving.

**80: Could you comment on the suitability of a Hybrid model to simulate PLISI and grounding line migration? How is this parameterized?**

There is no fundamental difference between hybrid SIA/SSA models, and higher-order / full-Stokes models, in their ability to simulate grounding-line dynamics, as the problems in doing so are caused by the discontinuity in basal friction at the grounding line, rather than by missing terms in the momentum balance. While the first MISMIP study (Pattyn et al., 2012; The Cryosphere) suggested that the full-Stokes model showed better results, this was because that model had a higher spatial resolution than the other ones. The need for the very high resolutions suggested in that study has since been negated by other modelling techniques, such as the flux condition scheme and the sub-grid friction scaling scheme.

IMAU-ICE uses a sub-grid friction scaling scheme to achieve good grounding-line dynamics at relatively coarse resolutions, similar to e.g., CISM (Leguy et al., 2014; The Cryosphere) and PISM (Feldmann et al., 2014; Journal of Glaciology). The IMAU-ICE model description paper (Berends et al., 2022; Geoscientific Model Development)

showed that this enables IMAU-ICE to resolve the migrating grounding line to within a single grid cell. We will add several sentences explaining this in more detail.

**89: In some ways the lacustrine environment might be quite different from the marine environment; the thermal structure may be different, and lakes could become chocked with icebergs. From a practical modelling perspective, it is reasonable that you treat marine and lacustrine the same, but you could discuss this further in the text.**

Indeed, there are many differences between the lacustrine and marine environment, which affect BMB and calving. For example, lacustrine calving is thought to be at least one magnitude smaller compared to tide water glaciers (e.g., Warren et al., 1995, https://doi.org/10.3189/S0260305500015998; Warrant and Kirkbride, 2003, https://doi.org/10.3189/172756403781816446). Thermal circulation is also different. In the ocean, circulation is driven by temperature and salinity gradient. The salinity gradient is absent in fresh water.

We will mention these differences in the introduction and discussion sections.

**93: Do you know (or could you know with some offline calculations) what proportion of the ice sheet margin is missing lakes because the model cannot simulate above sea level lakes?**

While technically possible, calculating the missing lakes is not trivial and computationally heavy.

The water level of a lake can be defined as the level at which water would start to flow towards the ocean. Therefore, it is important to be able to resolve smaller channels and valleys, as these determine lake levels. This requires e.g., a high topographic resolution, as lower topographic resolution can smooth out these valleys.

Therefore, we propose an alternative experiment to quantify the effect of this simplification. We selected a large region in North America where the water level of potential proglacial lakes where set to 50m above present-day sea level.

The results are shown below compared to Zero BMB and Rough Water simulation. The Lake 50m experiment loses more mass during interstadial periods, but is very similar compared to the Baseline during the deglaciation. A most, the ice volume during deglaciation in Lake 50m is a few centuries ahead of the Baseline. This effect is small compared to the Faster GIA and Rough Water experiments.

[Figure]

**101: merge > merged**

This mistake will be fixed.

**108: Not unfeasible! Millennial scale coupled climate-ice sheet simulation studies do exist, but it is understandable why this is not a reasonable modelling choice here. Please clarify.**

It is possible to run multiple millennial scale simulations with enough computational resources and time. We will change this in manuscript to explain that it is technically possible to run coupled climate-ice simulations, but at a large computational cost.

**126: The use of brackets to describe the reverse behaviour is a tad tricky to follow.**

We will change this sentence so it does not use brackets. Any other sentences in the manuscript that uses brackets will also be changed.

**164: A set 30% adjustment assumes that distal ice sheets fluctuate in unison with simulated ice sheets. Is this reasonable?**

The 30% addition to ice volume represents sea level change that does not result directly from North America, Eurasia and Greenland. These three ice-sheets

contribute around 100 meters in sea level decrease at LGM. (e.g., Simms et al., 2019).

In our study, we compare our results to eustatic sea level reconstructions. However, since we do not model all ice or sea level contributions (SLC), we need to add ~30% in order to compare our results directly to sea level reconstructions.

This is not perfect. Every 1 m sea level equivalent change in the Northern Hemisphere ice sheet volume does not necessarily equate to 30 cm additional sea level contribution from other sources.

Though, it should also be noted that Antarctic sea level contribution may be strongly correlated to Northern Hemisphere ice volume change. A sea level drop around Antarctica may prompt a grounding-line advance, which leads to ice volume increases (e.g., Gomez et al., 2020; Nature). A substantial part of the missing sea level may therefore be directly correlated to the modelled ice volume. Additionally, Northern Hemisphere ice volume is strongly correlated to the global temperature and consequentially the density of sea water and volume of smaller glaciers.

As a result, we will address that the 30% added to the ice volume does not perfectly represent the "missing" sea level change. However, it represents a rough estimate of the sea level change that we are not capturing with our model. Additionally, we will refer to the Gomez et al., 2020 paper to show that Antarctic volume and sea level change are correlated.

**Figure 4: This figure is challenging to follow, particularly panels b, d, and f. Are the points of deglaciation onset numerically defined? The onset of glaciation curves are difficult to read; I would suggest either removing or replotting. Ideally we shouldn't need a paragraph (lines 172-176) just to describe how to read a figure, not yet describing the results or discussion the implications.**

Figure 4 will be significantly reworked.

1) The blue "onset of glaciation" points will be removed. The climate that is needed to start a glacial cycle is not relevant to the overall story presented here.

2) The red "onset of deglaciation" points will be altered. These points are relevant enough to keep in the manuscript. However, it currently does not reflect the actual onset of deglaciations.

   Instead, we will now place an "onset of deglaciation" point when the ice sheet melts at a large enough volume (at least 20% of the modelled Late Pleistocene maximum) and has melted enough ice (less than 20% of the maximum Late Pleistocene volume remaining). These thresholds will be added to the caption.

3) External forcing index will be replaced by a simple "Glacial" and "Interglacial" climate to make it easier to understand.
4) Figure 7 (a similar figure) will be removed and replaced by 2D ice thickness maps.

The main goal of this figure is to show the difference in sensitivity between Eurasia and North America. And show that more glacial climates are needed to prevent an ice sheet to melt.

An updated version of the figure can be found below:

[Figure]

**190: Can you comment on the mechanism for the higher sensitivity of the Eurasian ice sheet?**

In the manuscript we have stated that the Eurasian ice sheet is more likely to melt during climate optima compared to the North American ice sheet.

Eurasia is thinner and smaller compared to North America, making Eurasia more likely to melt during climate optima. This is in line with one of the theories from the MPT (e.g., see Berends et al., 2021; Reviews of Geophysics). A small ice sheet (Eurasia), and large ice sheet (North America at LGM) are more likely to melt at climate optimum compared to a medium-sized ice sheet (North America at interstadial). Ice sheets maintain their own cold climate due to ice-albedo and temperature-elevation feedbacks, that may only compensate the climatic effect of an insolation maximum when an ice sheet is at least medium-sized. However, when

the ice sheet is too large, bedrock mass balance feedbacks, calving and large proglacial lakes make a large ice sheet more vulnerable to collapse.

The Eurasian ice sheet summer temperatures at LGM are also expected to have been higher (e.g., PMIP3 and PMIP4 LGM temperatures). A smaller increase in temperature may therefore yield a collapse of the Fenno-Scandinavian ice dome.

These processes will be discussed in the revised version of the manuscript. Additionally, this explanation will benefit from the newly added 2D maps of ice thickness and bedrock topography during the last deglaciation.

**211: The importance of the simulated ice shelves may depend on their spatial extent and if they are constrained laterally. It would be good to see a figure of simulated ice sheet location and morphometry.**

We will add several 2D maps of the last deglaciation showing ice thickness and bedrock topography during the last deglaciations. An example of this map is shown below:

[Figure]

**216: I don't follow the logic here?**

In the current version of the manuscript, we have made an attempt at explaining the Rough Water simulation.

To improve the explanation of the experiment, we will make changes in both the method section and results section (around line 216).

The Rough Water experiment benefits from an improved explanation of the sub-grid friction scaling scheme in the method section. In the Baseline simulation, friction is multiplied by the grounded fraction of the grid-cell. Therefore, basal friction is 0 for fully floating ice, and is reduced for partially floating ice (at the grounding line).

This will benefit the explanation of the Rough Water simulation, which will also be improved. In the Rough Water simulation, we do not multiply friction with the grounded fraction. Hence, basal friction is not decreased for floating ice, and is therefore treated as if all ice is grounded.

**Discussions: This section is very limited, it could be incorporated into the Results section.**

The discussion section will be expanded in the revised version of the manuscript. Therefore, it does not need to be incorporated in the results or conclusion sections.

**315: It would be good to develop this point a little further. What are the potential effects of lakes on future Greenland? And do models represent this?**

This would be an interesting concluding section to the paper. We will add a very brief discussion on 1) the vulnerability of the Western-Antarctic ice sheet with respect to MISI and 2) discuss the Greenland proglacial lakes, as these lakes may potentially accelerate Greenland melt in the future.

However, while there are analogies between past and future ice sheets, the Greenland and Antarctic ice sheets are very different from the North American and Eurasian ice sheets (for example for Eurasia, see van Aalderen et al., 2023; https://doi.org/10.5194/egusphere-2023-34 ). Antarctica SMB is low, and mass loss is dominated by basal melt and grounding line dynamics. Greenland is also much smaller compared to the North American and Eurasian ice sheets.

**423: It would be preferable for the simulations to be reproducible without contacting the author.**

We will add a data-acknowledgement section. To perform IMAU-ICE simulations, information on the initial bedrock topography, prescribed $CO_2$ (Bereiter et al., 2018) insolation (Laskar et al., 2004), climate (PMIP3) is needed. We are not the legal owners of these data-sets and we therefore cannot place these on a public database

without permission. However, we are allowed to create a data-acknowledgement section that provides the urls and references to the necessary data-sets.

---

## Author Response (AR1)

**Author reply**

First of all, we would like to thank our reviewers and editor. This document lists the changes we made for the revised version. Below the reviewers' comments are shown in bold, while our response is listed in regular font type. Line numbers correspond to the revised version of the manuscript.

First of all, we have re-run the simulations with maps of till friction angle based on measurements (North America; Gowan et al., 2019 (Nature), and Eurasia; Laske and Masters,1997 (EOS)) based on a request by the editor.

The main conclusions of the paper have not changed by this change in the set-up, though in the Rough Water simulation the Eurasian ice sheet can now fully melt during some interglacial periods. These changes improved in general the glacial inception periods, but in MIS13 there is not enough melt in the Baseline. The manuscript and its figures have been changed to accommodate the results of these new simulations.

The figure below is a comparison of the global mean sea level between the submitted and revised results of the original Baseline (dashed line) and our re-run Baseline (solid line):

[Figure]

**Response to the review by Niall Gandy**

We thank the reviewer for their helpful comments, and would hereby like to address the concerns they raised.

**Main points**

**Experimental design: I appreciate the novel approach of the experimental design, where you have made certain adjustments to better explore the ice sheet behaviour. However, the detailed justification for this is not explained sufficiently in the text, and I am left not fully convinced that your experiments explore the behaviour you intend. For example, the "Rough Water" experiment is designed to show the effect of negligible friction beneath ice shelves, but various feedbacks (surface profile, buttressing effects, a different GIA response to more grounded ice, ect) could be confusing the results. The experimental ethos could be explained in more detail.**

We hope to address these concerns with the following changes:

- We have expanded the model description in section 2.1 (Line 99-135). The method to determine the location of proglacial lakes is explained in more detail (see Line 122-125). The sub-grid friction scaling scheme, which we use to model the decrease in friction at the grounding line and floating ice, has also been explained in more detail as well (see Line 110-115). Equations that govern the basal roughness, basal hydrology, and the sliding law, have been added to a new appendix section (Line 491-525).

- We have also improved the description of the Rough Water simulation (see Line 294-299). This is also improved by the more detailed description in the method section of the sub-grid friction scaling scheme. In the Rough Water simulation, the sub-grid friction scaling does not differentiate between floating or grounded ice, but instead friction is treated as if all ice – including shelves –is grounded.

- We have conducted a few additional sensitivity experiments with basal friction. We have included a till friction angle map based on the geology map by Gowan et al. (2019; Nature) and Laske and Masters (1997; EOS). We have conducted two friction perturbation experiments where we apply homogeneous till friction angles of 10 degrees (roughly representing sediment cover) and 30 degrees (roughly representing bedrock). The results of these experiments are described in section 3.2 (Line 245-275). We found

that these perturbations in till friction angle have a strong effect on the deglaciation, with lower friction resulting in a faster deglaciation, and that friction has a larger effect on ice volume rather than ice area.

Additionally, we have also conducted two friction perturbation experiments for the Fast GIA and Rough Water to test if our main conclusions are consistent under different basal drag formulations. Fast GIA has the slowest melt followed by the Rough Water regardless of the till friction angle. Though the length of the deglaciation and the ice volume remaining during interglacial periods does increase with higher friction values. These experiments have been added to the supplementary information.

- We have added several 2D maps of the modelled ice-sheet geometry and the location of proglacial lakes. We show maps of different experiments during the last deglaciation which more clearly shows the differences between the experiments (see figures 3,7,9).

**You could also undertake some offline ice shelf mass budgeting (at each timestep what is the flow over the grounding line, what is lost to surface melt, sub shelf melt, and calving) to disentangle the behaviour.**

We have added a figure in the supplementary information to show the mean and range of the integrated mass balance, surface mass balance, basal mass balance and calving flux for the Baseline, Zero BMB, Rough Water, Low and High Friction, Fast and Slow GIA during terminations.

**Specific points**

**18: linger > remain?**

This word indeed fits better and has been changed in the manuscript (line 18).

**67: I think it is common to conflate susceptibility to MISI/PLISI and sub-shelf melting/calving. Could you clarify the mechanistic difference for the reader before this sentence?**

We have added a few sentences at line 60-63 explaining sub-shelf melting and calving. This is before MISI is explained, and should help to prevent confusion between PLISI and sub-shelf melting/calving.

**80: Could you comment on the suitability of a Hybrid model to simulate PLISI and grounding line migration? How is this parameterized?**

There is no fundamental difference between hybrid SIA/SSA models, and higher-order / full-Stokes models, in their ability to simulate grounding-line dynamics, as

the problems in doing so are caused by the discontinuity in basal friction at the grounding line, rather than by missing terms in the momentum balance. While the first MISMIP study (Pattyn et al., 2012; The Cryosphere) suggested that the full-Stokes model showed better results, this was because that model had a higher spatial resolution than the other ones. The need for the very high resolutions suggested in that study has since been negated by other modelling techniques, such as the flux condition scheme and the sub-grid friction scaling scheme.

IMAU-ICE uses a sub-grid friction scaling scheme to achieve good grounding-line dynamics at relatively coarse resolutions, similar to e.g., CISM (Leguy et al., 2014; The Cryosphere) and PISM (Feldmann et al., 2014; Journal of Glaciology). The IMAU-ICE model description paper (Berends et al., 2022; Geoscientific Model Development) showed that this enables IMAU-ICE to resolve the migrating grounding line to within a single grid cell. We have added several sentences to the method section explaining the sub-grid friction scaling scheme in more detail, and highlight its ability to model grounding line migration (line 107-114).

**89: In some ways the lacustrine environment might be quite different from the marine environment; the thermal structure may be different, and lakes could become chocked with icebergs. From a practical modelling perspective, it is reasonable that you treat marine and lacustrine the same, but you could discuss this further in the text.**

Indeed, there are many differences between the lacustrine and marine environment, which affect BMB and calving. For example, lacustrine calving is thought to be at least one magnitude smaller compared to tide water glaciers (e.g., Warren et al., 1995, https://doi.org/10.3189/S0260305500015998; Warrant and Kirkbride, 2003, https://doi.org/10.3189/172756403781816446). Thermal circulation is also different. In the ocean, circulation is driven by temperature and salinity gradient. This salinity gradient is absent in fresh water.

We have added a paragraph on these processes in the introduction section (Line 79-81) and discuss the uncertainty this may pose to our results (line 391-394).

**93: Do you know (or could you know with some offline calculations) what proportion of the ice sheet margin is missing lakes because the model cannot simulate above sea level lakes?**

While technically possible, calculating the missing lakes is not trivial and requires high resolutions (see Berends et al., 2016; https://doi.org/10.5194/gmd-9-4451-2016). The water level of a lake can be defined as the level at which water would start to flow towards the ocean. Therefore, it is important to be able to resolve smaller channels and valleys, as these determine lake levels. This requires e.g., a high topographic resolution, as lower topographic resolution can smooth out these valleys (now briefly discussed in lines 325-327).

Therefore, we have conducted alternative experiments to quantify the effect of the lake level. We simulated the last deglaciation with varying lake levels (100 m, 200 m, 300 m, 400 m, with respect to present day. As the lakes are placed where bedrock is below the lake level, this method allows the ice sheet to border higher lake levels. Though, this can also lead to inland oceans inhibiting ice inception, for this reason we only conduct these sensitivity experiments over the last deglaciation. Results show that the higher lake levels does lead to substantially faster deglaciation and this is both discussed in a new results paragraph (Line 320-336) and the discussion (Line 389-391). Figure 11 was also added to shows these results.

**101: merge > merged**

This mistake has now been fixed.

**108: Not unfeasible! Millennial scale coupled climate-ice sheet simulation studies do exist, but it is understandable why this is not a reasonable modelling choice here. Please clarify.**

It is indeed technically possible to run multiple millennial scale simulations with enough computational resources and time. We have changed this in manuscript to explain that it is technically possible to run coupled general circulation simulations, but at a large computational cost (see line 425-430). Alternatively, it is possible to run simulations on glacial time-scales with intermediate complexity models, though at the expense of the detail in the physical processes which can be captured as well as the spatial resolution in the model.

**126: The use of brackets to describe the reverse behaviour is a tad tricky to follow.**

We have changed this sentence so it does not use brackets. Any other sentences in the manuscript that uses this writing techniques have also been changed.

**164: A set 30% adjustment assumes that distal ice sheets fluctuate in unison with simulated ice sheets. Is this reasonable?**

The 30% addition to ice volume represents sea level change that does not result directly from North America, Eurasia and Greenland. These three ice-sheets contribute around 100 meters in sea level decrease at LGM. (e.g., Simms et al., 2019).

In our study, we compare our results to global mean sea level reconstructions. However, since we do not model all ice sheets and all processes that can amount to

sea level changes, we need to add ~30% in order to compare our results directly to sea level reconstructions.

This is obviously not perfect. Every 1 m sea level equivalent change in the Northern Hemisphere ice sheet volume does not necessarily equate to exactly 30 cm contribution from other sources and indeed the timing is also not necessarily coinciding with the NH ice sheets.

However, it is also not completely unreasonable. It should be noted that Antarctic Sea level contribution may be strongly correlated to Northern Hemisphere ice volume change. Sea level decrease around Antarctica may prompt a grounding-line advance, which leads to ice volume increases (e.g., Gomez et al., 2020; Nature). A substantial part of our missing sea level change may therefore be directly correlated to the modelled Northern Hemisphere ice volume changes, possibly with a small lag.

Therefore, we have addressed in the main text (see line 196-201) that the 30% addition to ice volume does not perfectly represent the sea level change that is not explicitly computed with our current set-up. However, it represents a first-order estimate of the sea level change that is not modelled here. Additionally, we have referred to the Gomez et al. (2020) paper to show that Antarctic volume and Northern Hemisphere induced sea level change are correlated.

**Figure 4: This figure is challenging to follow, particularly panels b, d, and f. Are the points of deglaciation onset numerically defined? The onset of glaciation curves are difficult to read; I would suggest either removing or replotting. Ideally we shouldn't need a paragraph (lines 172-176) just to describe how to read a figure, not yet describing the results or discussion the implications.**

Figure 4 has been significantly reworked to make sure it is easier to understand, and uses better thresholds.

1) The blue "onset of glaciation" points has been removed. The climate that is needed to start a glacial cycle is not relevant to the overall story presented here.

2) The red "onset of deglaciation" points has been changed. These points are relevant enough to keep in the manuscript. However, it did not reflect the onset of deglaciations accurately enough.

   Instead, we have placed an "onset of deglaciation" point when the ice sheet melts at a large enough volume (at least 30% of the modelled Late Pleistocene maximum) and has melted enough ice (less than 30% of the

maximum Late Pleistocene volume remaining). These thresholds have been added to the caption.

3) External forcing index has been replaced by a simple "Glacial" and "Interglacial" at the corresponding points in the graph. We hope this makes it easier for understand the graph.

4) Figure 7 (a similar figure) has been removed and replaced by 2D ice thickness map.

The main goal of this figure is to show the difference in sensitivity between Eurasia and North America. And to show that the climate needs to be colder/more glacial to prevent an ice sheet from melting or even collapsing. See figure 5 in the revised manuscript.

**190: Can you comment on the mechanism for the higher sensitivity of the Eurasian ice sheet?**

In the manuscript we have stated that the Eurasian ice sheet is more likely to melt during climate optima compared to the North American ice sheet.

Eurasia is thinner and smaller compared to North America making it more likely to melt during climate optima. Since the Eurasian ice sheet is thinner summer temperatures at LGM are also expected to have been higher (e.g., PMIP3 and PMIP4 LGM temperatures).  This is now discussed in lines 229-231.

A smaller increase in temperature may therefore yield a collapse of the Fenno-Scandinavian ice dome. This is in line with one of the theories from the MPT (Mid Pleistocene Transition; e.g., see Berends et al., 2021; Reviews of Geophysics). A small ice sheet (Eurasia), and large ice sheet (North America at LGM) are more likely to melt at climate optimum compared to a medium-sized ice sheet (North America at interstadial). Ice sheets maintain their own cold climate due to ice-albedo and temperature-elevation feedbacks, that may only compensate the climatic effect of an insolation maximum when an ice sheet is at least medium-sized. However, when the ice sheet is too large, bedrock mass balance feedbacks, calving and large proglacial lakes make a large ice sheet more vulnerable to collapse. This is discussed in lines 39-49.

**211: The importance of the simulated ice shelves may depend on their spatial extent and if they are constrained laterally. It would be good to see a figure of simulated ice sheet location and morphometry.**

We added several 2D maps of the last deglaciation showing ice thickness, lakes, and bedrock topography during the last deglaciation. These can be seen in figures 3,7 and 9.

**216: I don't follow the logic here?**

Here we attempted to explain the Rough Water simulation. Though we have now implemented some changes to the manuscript to improve this explanation.

The improvements are made in both the method and results section. We have improved the methods to include a better explanation for the sub-grid friction scaling scheme, which is at the base of the change we made for the Rough Water simulation: In the Baseline simulation, friction is multiplied by the grounded fraction of the grid-cell. Therefore, basal friction is 0 for fully floating ice, and is reduced for partially floating ice (at the grounding line). This can be seen in lines 110-115.

We have also improved the description for the set-up of the Rough Water experiment in the results. In the Rough Water simulation, we do not multiply friction with the grounded fraction. Hence, basal friction is not decreased for floating ice, and is therefore treated as if all ice is grounded. This can be seen in lines 294-300.

**Discussions: This section is very limited, it could be incorporated into the Results section.**

Several paragraphs have been added to the discussion in the revised manuscript. Therefore, we believe there is no longer a need to incorporate it with either the results or conclusion sections.

**315: It would be good to develop this point a little further. What are the potential effects of lakes on future Greenland? And do models represent this?**

This made an interesting concluding section to the paper. We have added a very brief discussion on 1) the vulnerability of the Western-Antarctic ice sheet with respect to MISI and 2) the Greenland proglacial lakes, as these lakes may potentially accelerate Greenland melt in the future.

However, while there are analogies between past and future ice sheets, we have also stated that Greenland and Antarctic ice sheets are very different from the North American and Eurasian ice sheets. Antarctica SMB is low, and mass loss is dominated by basal melt and grounding line dynamics. Greenland is also much smaller compared to the North American and Eurasian ice sheets. See lines 456-464.

**423: It would be preferable for the simulations to be reproducible without contacting the author.**

We have added a data section in lines 484-490, and also refer to the GitHub page of IMAU-ICE. To perform IMAU-ICE simulations, information on the initial bedrock topography, prescribed $CO_2$ (Bereiter et al., 2018) insolation (Laskar et al., 2004), climate (PMIP3) is needed. We are not the legal owners of these data-sets and we therefore cannot place these on a public database without permission, but we have now provided the relevant urls and references so people can find the input data.

**Response to the review by Fuyuki Saito**

We thank the reviewer for constructive and helpful commentaries. We would hereby like to address the concerns raised. Reviewer comments are shown in bold and our responses are shown in regular font type.

**Description of the ice-sheet model: Section 2.1 describes the ice-sheet model briefly, but I feel it too short. A complete repeat of the previous papers are not necessary, but at least the description of the methods relating to the sensitivity experiments of the present paper should be included in detail.**

We have improved the model description section (lines 99-136) based on the suggestions listed below (treatment of grounding line migration, modelling of proglacial lakes, and basal sliding).

Additionally, we have added an appendix (491-524) which describes the equations governing till friction angle, basal hydrology and resulting basal friction.

**(i) how to deal the grounding line migration in the model. it is all right without any special treatment, but mention explicitly.**

We use a sub-grid friction scaling scheme to achieve good grounding-line dynamics at relatively coarse resolutions, similar to e.g., CISM (Leguy et al., 2014; The Cryosphere) and PISM (Feldmann et al., 2014; Journal of Glaciology). The IMAU-ICE model description paper (Berends et al., 2022; Geoscientific Model Development) showed that this enables IMAU-ICE to resolve the migrating grounding line to within a single grid cell. We added several sentences at line 110-115 explaining the sub-grid friction grounding scheme in more detail.

**(ii) how to deal the formation of proglacial lakes. I suppose that the depth of lake is the difference between the background sea level and the bedrock.**

This is correct. We have added a few sentences to explain this method, and justify our choice for this simplification. See lines 124-127.

**(iii) explicit equation of basal sliding, and the difference in how to apply between base-line and rough-water experiments.**

The equations governing the basal roughness, basal hydrology, and the sliding law, have been added in a new appendix (lines 491-524). We have also included a more thorough description of the Rough Water simulation (see lines 294-300), in which the basal friction is treated the same regardless if the ice is floating or not.

**Other points:**

**Abstract: There are no description of the method used in the present paper, even no explanation that the discussion is based on a series of numerical simulation. This must be clarified.**

This is indeed an oversight in the abstract. We have added a few sentences at line 13 and 14 in the abstract to describe the types of experiment that were conducted. These include 1) explicitly mention we have conducted a numerical modelling study and 2) that we varied the representation of certain processes in this model to determine the effects of GIA, ice-dynamics, friction under grounded or floating ice, and basal melt.

**BMB experiments: Typical mass balance terms, both surface and base, over ice shelves in the baseline experiment should be mentioned, which will help to get typical ratio of surface/basal mass balance terms.**

We have added figures comparing mass balance, surface mass balance, basal mass balance, calving and ice volume to the supplementary information.

**Figure 2. Plotting two points corresponds to PI and LGM may help.**

We have added a point for LGM and PI to figure 2. We have tuned the insolation to be able to simulate glacial cycle periodicity and obtain a reasonable LGM volume and melt at present-day. Though, since PI and LGM have different insolation at 65N, at 280 ppm $CO_2$ and PI insolation, which is also combined with the dependency of climate forcing to albedo, these points will be close to, but not exactly match, present-day conditions.

**Figure 3. Better to mention in the caption that this is the result of baseline experiment.**

We have added "Baseline" to the caption to show that figure 3 indeed shows the Baseline experiment.

**Figure 4 and main text. Please note the definition of the onsets of (de)glaciation in this paper. Why some points (e.g., 240ka in the total and Eurasia, 290ka in North America) are not detected as the onsets?**

The onset and termination points were based on a few thresholds. First of all, the ice sheet needed to be big enough and melt for a long enough duration. However, we have decided to change these thresholds and rework the figure.

The following changes have been made to improve figure 5.

1) The red "onset of deglaciation" points are now located at the right location (only at glacial maxima).

An "onset of deglaciation" point is added when the ice sheet melts at a large enough volume (at least 30% of the modelled Late Pleistocene maximum) and has melted enough ice (less than 30% of the maximum Late Pleistocene volume remaining). These thresholds have been added to the caption.

2) The blue "onset of glaciation" points has been removed as they added little to the overall story while making the figure more complicated.

3) "External forcing index" is replaced by "glacial" and "interglacial" climate to make the figure easier to understand.

4) Figure 7 (a similar figure) has been removed and replaced by 2D ice thickness maps.

**Appendix Eq.(1) Please clarify that f_{snow} is no more than 1.  I calculated by hand, and observed that f_snow become 1 when T - T0 is around -10K, and exceeds 1 when -11K.  I am not sure my rough computation is correct, but please clarify the actual computation of f_{snow} (Of course it is all right if T - T0 is always more than -10K).**

f_snow in the model is limited between 0 and 1, though this was not stated in the manuscript. Therefore, f_snow will not exceed 1 even when temperatures are well below 262K (see equation 22). We have added a sentence at line 605 clarifying that f_snow is limited between 0 and 1. When other parameters have such limits, we made sure to now mention these in the manuscript.

**Melt_{prev}.  Again, it is hard to check whether the albedo is between 0 and 1 by this equation.**

In line 617 we now state here that the albedo is limited between the background albedo (depending whether the surface is ocean / land / ice) and the albedo of fresh snow.

**Eq (4). Need units of the coefficient 0.012.**

Units have now been added to equation 25.

**Eq (21).  Need unit(scale) of P_{PI,corr} to compute log.  Or, is the first term e means exp() function, not e times ()?**

Equation 21 (now 13), should indeed have been exp, and we changed it accordingly. A similar change has been made in equation 34.

**Response to the editor comments by Lev Tarasov,**

We thank the editor for constructive and helpful commentary and would hereby like to address some of the issues they raised.

**As you draft your response to reviewers, I have identified 4 other major issues that need to be addressed.**

   **1) Experimental design.**

**The current design leaves too many uncertainties unaddressed in part stemming from model limitations somewhat buried or not even described in your much too brief model description (as already raised by one of your reviewers).**

The model description has been expanded (see lines 99-136). There are three processes that have now been explained in more detail:

1) Treatment of proglacial lakes (lines 124-128). We use the difference between bedrock and sea level to determine the location of lakes. If sea levels exceed bedrock topography (and there is no grounded ice), it is considered a lake. We also acknowledge that as a limitation in line 389-291.
2) Basal hydrology was not mentioned in the method, and this should have been there from the start. We briefly explain the basal hydrology method in lines 103-105. An appendix has been added that explains how basal hydrology, basal roughness and sliding law are implemented in our model and presents the governing equations. This can be found in lines 491-524.
3) The treatment of grounding line migration. Here we used a sub-grid friction scaling scheme. This process is important to all our experiments, so we have explained it in much more detail (see lines 107-114).
4) We have also re-run all simulations using till friction angle maps based on (from Gowan et al., 2022 (North America) and Laske et al., 1997 (Eurasia)), which is also explained in the method section (see lines 104-106).

We have also conducted high and low friction cases for Baseline, Fast GIA, and Rough Water to show that our main conclusions are consistent regardless of basal friction. These simulations have been conducted by changing the friction mask to 10 degrees (roughly representing a full sediment coverage) and 30 degrees (roughly representing a full bedrock coverage). We believe these are the two most extreme till friction angle scenarios. Though note that the basal hydrology method does not distinguish between sediment or bedrock coverage, nor is there sediment transport. Therefore, the analogy does not fully hold.

The Low Friction / High Friction experiments are added to a new paragraph (section 3.2; lines 245-274) in the Results section. The Low Friction / High Friction equivalent

of the Fast GIA and Rough Water simulations have been added to the supplementary information.

We furthermore discuss this limitation in more detail in the discussion section, as well as indicate that there are more sophisticated methods to deal with basal friction (e.g., sediment transport models) and basal hydrology (e.g., distinguishing between sediment and bedrock). For this, see lines 410-412.

**1a)**

**As far as I can tell, you do not use a soft/hard bed mask for Eurasia, even though it has long been known that this can have a large impact on ice sheet response and geometry (eg cf Tarasov and Peltier, 2004). So a significant fraction of your sensitivity to basal drag changes from pro-glacial lakes could be due to this easy to address design limitation as regions that are soft-bedded will generally have less basal drag (when warm-based).**

Originally, we used a parameterization from Martin et al. (2011). However, when we needed to re-run the simulations, we decided to include a till-friction angle map based on observations. Here we use the geology mask from Gowan et al. (2019), which is made specifically for ice-sheet modelling. For Eurasia, we have used the sediment thickness from Laske et al. (1997), where sediment thicknesses exceeding 100 m receive a till friction angle of 10 degrees, and thinner sediment regions receive 30 degrees. These till friction angle maps are static – there is no sediment transport – and we apply the same basal hydrology scheme to both. These limitations are described in the discussion section at line 408-413.

Additionally, to show the sensitivity of the basal drag, we have replaced the till friction angle maps with one of 10 degrees (Low Friction simulation) and 30 degrees (High Friction simulation). A new paragraph has been written to show the results of these two additional experiments (see lines 245-274). These include showing a time-series as well as 2D ice thickness / bedrock topography maps of the last deglaciation. Though it should be noted that these situations do not represent the exact difference between sediment and no-sediment strata. Instead, it shows that glacial cycles and particularly deglaciations are sensitive to perturbations in the basal friction. It therefore shows a considerable sensitivity to the basal friction treatment.

**1b)**

**I suspect you have no basal hydrology (again model description is way to brief on relevant aspects of the model configuration) and**

**I also suspect you are simply taking the adjacent water depth to compute effective pressure for the Budd sliding law. On a 40km grid, that choice is hard to defend. At the very least, try an easy to implement leaky bucket approach (**

**eg as in PISM or https://tc.copernicus.org/preprints/tc-2022-226/) for comparison.**

Our basal hydrology parameterisation is based on Martin et al., 2011 (The Cryosphere), and has also been used in PISM. Here, the pore water pressure is calculated based on the ice thickness, bedrock topography and sea level.  This is now described briefly in the methods (see lines 103-104).

We have added a description of basal hydrology to the manuscript, as well as a new appendix describing the basal hydrology, till friction angle and resulting basal friction and sliding law. See lines 491-524.

**1c) The assumption that lake depth is given by bed depth below contemporaneous sealevel is another major limitation, eg long known in the geological community (or cf Tarasov and Peltier 2006 from a modelling perspective) from the critical dependence on controlling sill elevation for eg Laurentide Lake Agassiz. Surface drainage solvers have been in some published models for almost two decades. I can understand adding one is non-trivial, but at the very least more thought needs to be put into addressing the some of the uncertainties arising from this model limitation.**

To address these issues, we have conducted a four additional experiment to quantify the effect of this simplification. We have conducted the last deglaciation with sea levels of 100 m, 200 m, 300 m and 400 m above present-day. Since lakes are placed where bedrock is below sea level, this causes the ice sheet to border increased lake levels relative to the Baseline simulation. We have only conducted these experiments over the last 21 thousand years –branching off the Baseline at the LGM – as the higher sea levels can create inland seas that inhibit inception and growth of the ice sheet.

These results are shown in section 3.5, lines 320-336 as well as figure 11.

**2) Scientific precision/accuracy/transparency**

**Eg:**

**"We find that the modelled sea level matches the reconstructions well"**

**Your model fails to capture any signal of last glacial inception, so blanket use of "matches the reconstructions well" is inaccurate.**

We have weakened this too strong statement (see line 205). Glacial inception is not the key focus in these simulations, and could be more addressed in more detail. Though, in the new simulations that include the till friction angle maps, the inception periods are improved.

**No where do you show your LGM ice extents (only 11 ka), instead your only LGM ice extent map shows a cited map from Abe-Ouchi et al (2015). This may suggest to the reader that you do not want them to see what our LGM ice extent looks like.**

We have added maps of the LGM ice sheet – as well as other time-slices – to the manuscript, and compare it to Abe-Ouchi et al. (2015) and Dalton et al. (2020). See figures 3, 7 and 9.

Our ice sheet extent at LGM matches reconstructions reasonably well, compared to other freely-evolving paleo-ice-sheet models (e.g., Berends et al., 2018. Willeit et al., 2019). Though, we lack substantial ice coverage in Scotland and Southern Alaska. This mismatch was not described in our results, but has now been added to section 3.1 (lines 185-187).

Though we do find that our ice sheets melt too fast compared to the reconstruction by Dalton et al. (2020). This discrepancy starts from ~12 ka, corresponding to the Younger Dryas, a cold period in the Northern Hemisphere often attributed to a stagnation in the AMOC. We believe that this discrepancy is mostly caused by the lack of a melt-water–ocean – climate feedback. As the large fresh water influx into the North Atlantic should prompt cooling in the Northern Hemisphere, perhaps contributing to the lower Younger Dryas melt rates observed in reconstruction (e.g., Lambeck et al., 2014 | PNAS). In our current set-up we do not obtain a cooling during the Younger Dryas as our temperature forcing is determined by $CO_2$, insolation and albedo. This limitation is now discussed in both the results (lines 187-194) and discussion section (lines 419-424).

**Or**

**"our simulations tend to have slightly too long interglacial periods compared to reconstructions" From what I can read off of your plot in figure 3, your last interglacial is @ 50kyr too long, that is not "slightly".**

Interglacial periods are too long in our simulations and we have now emphasised this better in the results and discussion section (lines 185-187 and lines 375-377) Though, while our interglacial periods still prompt too little ice growth, this discrepancy is improved in the re-run simulations.

**Adequate referencing of past relevant litterature, eg role of GIA in deglaciation of ice sheets (eg Tarasov and Peltier,
Ann. Glac., vol. 25, 58-65, 1997), impact of pro-glacial lakes on
ice extent (eg Cutler et al, 2001, Geology), or relative sensitivity
of Eurasian versus North American ice sheets to orbital/CO2 forcing (eg Tarasov and Peltier, JGR 1997).**

We have added these, and several other relevant references for these topics.

**4) Claims in conclusions that havxe not been shown, eg: "3.4 Glacial isostatic adjustment ... The North American ice sheet may not even fully deglaciate during some interglacial periods. This is because proglacial lakes are not created when the bedrock uplift is too fast". This is not shown in your results (which would require a further sensitivity experiment with lakes turned off for soft/hard GIA experiments).**

We have added 2D maps with time-slices from the last deglaciation comparing the Fast GIA, Slow GIA and Baseline simulations in figure 9. These figures show that proglacial lakes become smaller with increasing uplift rates. The maps of the Fast GIA simulation shows that the proglacial lakes are much smaller and mostly absent compared to the Baseline simulation.

Additionally, we conducted the Fast GIA, Rough Water and Baseline simulations with a low till friction angle map (10 degrees, representing full sediment coverage) and high friction (30 degrees, representing full bedrock coverage). While the till friction angle does have a substantial effect on the timing of deglaciation, it does not change the main conclusions of the paper: The Fast GIA has a slower deglaciation compared to the Rough Water experiments, and the Baseline has the fastest deglaciation regardless of till friction angle.

Results of these additional friction experiments using the Fast GIA and Rough Water have been added to the supplementary information. Similar friction experiments with the Baseline simulations are discussed in a newly added paragraph in the results section (lines 245-274).

**Furthermore, ice volume is a very limited metric. Lake calving will more directly affect ice extent than ice volume (though the two are obviously related). Without relevant map-plots (at least in a supplement), the reader is unable to adequately evaluate the extent to which you results match your claims.**

Indeed, we have now added 2D ice thickness maps to the manuscript showing the last deglaciation of the Rough Water, Baseline, Zero BMB, Fast GIA and Slow GIA, as well as the newly created Friction experiments. These maps show the Northern Hemisphere ice sheets at different time-periods (e.g., last glacial maximum, 11 ka, 9ka, present-day) and can be seen in figures 3,7 and 9. Additionally, we have added a time-series of ice volume and area supplementary information for the Baseline, Low Friction and High Friction.

*While technically possible, calculating the missing lakes is not trivial and computationally heavy.*

**Compared to the ice dynamics, the surface drainage and lake solution can be quite cheap (in the GSM (glacial systems model), it's less than 10% of overall GSM compute load).**

Calculating the lake solution is cheap when done at the same spatial resolution as the ice-dynamical model. However, Berends and van de Wal (2016) showed that this can lead to a significant overestimation of the water surface level and volume, as narrow outlet channels (e.g. river beds) are not resolved. They demonstrated that a resolution of 2 km is required to get accurate results, which drastically increases the computational load. This is now briefly discussed in lines 325-327.

*3)We have conducted a few additional sensitivity experiments. We have conducted the Baseline simulation with a respective 50% increase and decrease in the till friction angle*

**Ensure you justify your chosen sensitivity ranges, so that the reader can judge whether you've covered maximum plausible bounds, eg as a case in point:**

We have decided to change the set-up of these simulations. We have conducted each simulation with sediment / geology masks (Gowan et al., 2020 for North America, Laske et al., 1997 for Eurasia with a 100 m cutoff between sediment and rock). We have used the geology mask from Gowan et al. (2019) as it (1) has full coverage in our ice domain and gave us good results at the LGM. After this we retuned the model, which gave us much improved inception periods (though the growth of ice is still generally too little compared to reconstruction).

This also gave us the opportunity to change the Low Friction and High Friction experiments into something more tangible. Now, the Low Friction roughly represents a fully sediment covered domain, represented by a till friction angle of 10 degrees. The High Friction experiment changed into Bedrock, which has a till friction angle of 30 degrees across the entire domain.

We believe these exceed the minimum and maximum plausible bounds, and yields an interesting addition. These are discussed in lines 245-274.

We conducted these two friction perturbations for the Fast GIA and Rough Water as well. Though we have added these to the supplementary information.

*To address these issues, we propose an alternative experiment to quantify the effect of this simplification. We select a large region in North America (south of the Hudson Bay to southern margin of the domain) where the water level of potential proglacial lakes is set to 50m above present-day sea level.*

**Just choosing 50m is running blind. Refer to previous studies that explicitly model surface drainage to come up with appropriate upper and lower bounds for lake surface elevation relative sealevel.**

We have decided to change the experiment and only simulate the last deglaciation with sea levels of 100 m, 200 m, 300 m and 400 m. The lake levels likely changed substantially over time, and this experiment is to show that the lake levels can cause substantial increase of the melt rates. These are discussed in lines 320-336.

*>89: In some ways the lacustrine environment might be quite different from the marine environment;...*

*We will mention these differences in the introduction and discussion sections.*

**Just mentioning differences might not be enough. Experimental uncertainties that are easy to address (eg reducing calving and submarine melt coefficients to reasonable lower bounds) are best done or at least arguments to need be provided as to why they are not relevant.**

We have added a discussion on the uncertainty in the basal melt and calving rates. Due to large uncertainties in the basal melt of Lake Agassiz (lines 391-403), we have emphasized that the difference between the Baseline and Zero BMB should be taken as qualitative rather than quantitively (lines 399-403).

*We will change this in manuscript to explain that it is technically possible to run coupled climate-ice simulations, but at a large computational cost.*

**Depends on which climate model is used. For EMICs, this is more accurately moderately large cost.**

We have changed this in the manuscript. Earth system models of intermediate complexity are discussed and explained why it is a possible way to simulate glacial cycles with more explicit climate-ice sheet feedback, though at an increased cost relative to our set-up (see lines 463-464).

*Our ice sheet extent at LGM matches reconstructions reasonably well, compared to other freely-evolving paleo-ice-sheet models (e.g., Berends et al., 2018. Willeit et al., 2019).*

**Neither of the above make use of available geological constraints for LGM extent. For North American and Eurasia ice extent, geological reconstructions are more appropriate. Current most appropriate for comparison would therefore be Dalton, et al, QSR 2020. I can provide latlong rasterized**

**isochrones if you can't deal with the shapefiles attached to the above (lev@mun.ca).**

We have now added a map comparing the Baseline to the Dalton et al., 2020 extent (see figure 3). Our deglaciation is fast compared to Dalton from ~12 ka onwards. We believe this is due to climate forcing, as we lack the interactions between ocean and melt-water, and even our experiments with the lowest melt rates have smaller extent than Dalton at ~10-9 ka.

This discrepancy from ~12 ka roughly corresponds to the Younger Dryas, which is often attributed to a stagnation of the AMOC. Our model set-up lacks this interaction. We therefore miss the cooling that would have been caused by such a change in ocean dynamics. This limitation has now been described in the results (lines 187-194), and discussion section (lines 419-424).

---

## Author Response (AR2)

Response to the review by Fuyuki Saito:

First of all, we would like to thank the reviewer for these comments. We have made these minor changes to the manuscript, and ensured there is a clearer distinction between months and meters in the equations. To go into more detail (the reviewer comments in bold, and ours in regular font-type):

**Now I am very happy to see that all the concerns I raised at the first review are solved, except for some minor technical correction. Indeed, they are not the author's but the reviewer's (my) fault, about units in some equations.**

**Eq. (24) m is used as variable in the left hand side, thus the units m appended to two coefficients (-15, 0.015) are really confusing. I am very sorry to have suggested this, but it is better to remove. Instead, please mention D is expressed in terms of unit meter in the main text.**

We have removed the "m" in equation 24 and made sure to mention that both firn depth (D) and the amount of melt previous year (also Eq. 24) are in meters.

**Eq. (26) the same as above. Remove m/yr, and please define the unit of s term is m/yr.**

We now mentioned that the variables $s$, but also $I$, and $P$ have units of $m/yr$.

**In addition, please define m corresponds to month (I suppose) somewhere in the text.**

We have replaced every instance of m (as months) with $mnth$ (in equations).

Response to the review by Niall Gandy:

**Re-Review of "Late Pleistocene glacial terminations accelerated by proglacial lakes" by Scherrenberg et al.**

**Signed: Niall Gandy**

**Thank you for your careful and detailed response to my reviews of the previous draft of this manuscript. The revised manuscript is much improved, and it is clear that you have invested some considerable effort responding to reviewer and editor comments. Of specific note;**
1. **There are significant improvements in the justification of the experiment design. I find the new 2D map plots particularly helpful in understand the modelled behaviour. The improved model description and description of the "Rough Water" simulation is also helpful. Whilst not a material change to the science output, these changes are helpful for the readership.**
2. **The addition of the flux budgeting figure in the supplementary information is also helpful in understand the behaviour of the system.**
3. **The careful revision of Figure 4 is a significant improvement.**

**I note that there are some remaining questions from the editor on your response to my initial comments, including on the computational cost on calculating the missing lakes, on the choice of proglacial lake level at 50m, and on a clearer justification of the potential difference between lacustrine and marine dynamics. Assuming a robust response to these points, I would recommend this manuscript for publication.**

We would like to thank the reviewer for their comments, and we have now answered the remaining comments from the editor.

Response to the editor's comments by Lev Tarasov:

First of all, we thank the editor for their comments. Editor's comments are listed in bold, ours as regular font type.

**# Final points to address ('#' prefix denotes editor paragraph)**

**# ensure you have somewhere indicated your conversion factor for ice volume to sea level equivalent, and whether you account for changing ocean area and exclude floating ice in your sea level calculation.**

We have added a sentence to the method section to explain how we calculated sea level. While we do not account for a change in ocean area, we do exclude floating ice in our sea level calculation (see line 127)

**we generated a till friction angle map based on the sediment thickness map from Laske and Masters (1997), where we use till friction angles of 10 degrees for sediment thicknesses below 100 meters and 30 degrees for thicknesses exceeding that threshold.**

**# I assume you meant the reverse?**

Indeed, this was an oversight in the text and we used high friction in regions without sediment, and lower friction in regions with sediments. This was correctly mentioned in the appendix, but not the method's section. We have now corrected it.

**Also I find your 100 m threshold quite problematic as subglacial till deformation is mostly confined to the top few meters. A somewhat larger allowance can be made on the basis of topographic roughness, but I doubt the continental terrain of the Eurasian and North American ice complexes warrants 100 meter except for a few localized regions. You will need to justify the choice threshold in the paper (it may be that the coarse Laske and Masters will give about the same with eg a more defensible 20 m threshold), or at least provide a hindsight caveat.**

Indeed, the threshold we used is too large, though the difference between using 20 m and 100 m threshold is small in the Laske and Master's map as it has a very coarse resolution (both spatial and in terms of sediment thickness). We do not expect the 20 m and 100 m threshold to make a large difference as it effects only a small selection of grid-cells. The figure below (which will not be added to the manuscript) shows the difference between a 20 m and 100 m threshold:

[Figure]

[Figure]

Nevertheless, we have change this in the text and stated this as a possible oversight. For North America, we used a mask based on Gowan et al. (2019), which we preferred due to the higher resolution, and as it was specifically made with ice-sheet modelling in mind, but this does not cover Eurasia (line 111).

**interpolate between pre-calculated pre-industrial .. and forms a good alternative**

**to fully-coupled ice-climate set-ups**

**# On what basis do you justify "good"? On it's own, I do not see it as a good alternative in this day and age. If "good" means gives the results you want, than state that specifically. Best to avoid subjective claims and provide approximately quantitative or more informative qualitative descriptions. Leave the judgments to the reader.**

We have removed "good" from the text, and replaced the sentence:

*"This allows us to implicitly include climate – ice-sheet interactions at low computational costs and forms a good alternative to fully-coupled ice-climate set-ups, which have high computational costs"*

With:

*'This allows us to implicitly include climate – ice-sheet interactions at low computational costs compared to fully-coupled ice-climate set-ups.'*

(See line 147-148)

***The modelled LGM extent matches the reconstructions reasonably well, though ice coverage is lacking in the British island.***

**# unsupported claim, given missing EA deglacial reconstruction. Include the well-constrained DATED reconstruction of Eurasian deglacial ice margin retreat (Hughes et al, 2015, the first author can provide netcdf raster maps) for figure 3.**

We have now included the Hughes et al. (2015) maps to figure 3, and changed the caption of the figure.

**# more figure 3 points**
**1) Please add the present-day landmask and large lakes contours. It will help readers geo-locate and for instance see evolution of Hudson Bay relative to today.**

We have added the present-day coastline and contours showing larger lakes (e.g., the Great Lakes) to the figure as black contours. We have made this change not only in figure 3, but also in any figure showing 2D fields (Fig. 7, Fig. 9, and maps in the supplementary information).

**2) As your ice sheet has already deglaciated by 10ka, comparing to 5ka is not illuminating. Please replace the 5ka comparison with 8 ka, especially given your claim:**

We have replaced the 5 ka panel with 8 ka in figure 3.

**"with full retreat already reached at 10 kyr ago rather than two millennia later"**
**# this last statement is incorrect statement (deglaciation is not complete by 8 ka), which a plotted 8ka Dalton et al time-slice would make clear.**

This statement should indeed have been 3 to 4 millennia later instead. Full deglaciation is reached roughly between 7-6 ka, so ~3 to 4 millennia later. There is some ice volume loss after, but this retreat is small compared to the main phases of deglaciation. (Line 200).

***This is likely due to the absence of a feedback between melt water, the ocean and climate. For example, prior to the Younger Dryas (12.9-11.7 kyr ago),***

**# On the surface, this would only explain a max 1.3 kyr discrepancy (length of Younger Dryas), especially since AMOC resumption tends to entail overshoot. More to the point, the above claim implies that ignoring all the physics of eg atmospheric circulation is not important. The absence of a dynamically coupled ocean (or some representation thereof) is definitely one factor, but I see no basis for the implied claim that it is the only relevant factor.**

This is correct, it is one of the shortcomings, but there is multiple. We have changed this sentence to explain that (1) there could be multiple reasons for the discrepancy and (2) that the lack of ocean-ice interactions could partially (not entirely) explain it. (see line 201 and line 449).

**We find that the modelled sea level matches the reconstructions well**

**# Definitely not for last glacial inception. Provide a more precise/accurate claim This point was raised in my previous editor review, and your current response to the editor states: "We have weakened this too strong statement", but clearly this is not the case.**

We have removed this statement from the manuscript.

**Consequently, ice inception requires relatively low CO2 concentrations and weak insolation.**

**# insert "modelled" before "ice" since MIS5d (according to sea-level inference) is a clear counter example.**

Indeed, this should have been specified. We have changed this accordingly.

**Note that we apply these sea levels on the entire domain, as determining the exact location of lakes requires a high spatial resolution, as lower resolution can smoothen valleys and therefore miss drainage pathways (Berends et al., 2016).**

**# The only relevant meaning of "exact" here is with the ISM grid resolution. And the cited reference has not shown that a high spatial resolution is required to get a lake volume within say 5% of the upscaled high resolution calculation (and even 5% accuracy is over the top given all the other uncertainties and resulting errors in the model to do with climate forcing, basal drag, GIA,..). To show that, one would have to compare upscaled volumes to that calculated with a more appropriate choice of 40km bed topography that accounts for subgrid drainage channels (cf the already cited Tarasov and Peltier 2006). Furthermore, I strongly suspect that even running the code of Berends et al., 2016 at say a cheap 10km resolution every 100 years with uncorrected bed would still give much less error than obtains by the current very crude sea-level approach.**

Yes, this may provide a mid-way solution between a lake-model and computational resources. Since this deserves a broader explanation, we have moved the discussion on lake models from the results (line 340) to the discussion (see line 405-410) section.

**# Current wording is confusing. Do you actually raise sea-level by this amount, or just the geoid for lakes? Hopefully just later, in which case please phrase more clearly (eg "terrestrial geoid for lakes" or some such)**

Sea level was raised in the entire grid. We have now specified this further in lines 348.

**Appendix B: Climate time-slice interpolation**

**# how did you choose your relative weighting between CO2, insolation, and albedo?**
**Was this by adhoc tuning or by a specific methodology that led to**
**unique weights. Briefly spell out approach used, especially if latter.**

The albedo weighing (Eq. 6-7 and 11-12) are based on Berends et al. (2018). This is now specified in appendix B.

For the $CO_2$ vs insolation scaling (Eq. 8), we used a new approach: First we conducted a preliminary simulation based on the method by de Boer et al., 2012 ( https://doi.org/10.1007/s00382-012-1562-2) and Berends et al., 2021 (https://doi.org/10.5194/cp-17-361-2021) where external forcing is changed to match the modelled and observed benthic $\delta^{18}O$ record (Ahn et al., 2017). As such our model includes a routine to calculate $\delta^{18}O$ from ice and deep-water temperature, again based on the aforementioned works. This method essentially gave us the forcing needed to match the $\delta^{18}O$ record. We then fitted summer insolation and $CO_2$ to this curve, and we obtained Eq. 8. Since this method is beyond the scope and focus of this paper, we have briefly summarised it in Appendix B.